# EdiBERT: a generative model for image editing

**Thibaut Issenhuth**                                                   *t.issenhuth@criteo.com*
*Criteo AI Lab, Paris, France*
*LIGM, Ecole des Ponts, Univ Gustave Eiffel, CNRS, Marne-la-Vallée, France*

**Ugo Tanielian**                                                      *u.tanielian@criteo.com*
*Criteo AI Lab, Paris, France*

**Jérémie Mary**                                                        *j.mary@criteo.com*
*Criteo AI Lab, Paris, France*

**David Picard**                                                      *david.picard@enpc.fr*
*LIGM, Ecole des Ponts, Univ Gustave Eiffel, CNRS, Marne-la-Vallée, France*

**Reviewed on OpenReview:** *https://openreview.net/forum?id=GRBbtkW3Lp*

## Abstract

Advances in computer vision are pushing the limits of image manipulation, with generative models sampling highly-realistic detailed images on various tasks. However, a specialized model is often developed and trained for each specific task, even though many image edition tasks share similarities. In denoising, inpainting, or image compositing, one always aims at generating a realistic image from a low-quality one. In this paper, we aim at making a step towards a unified approach for image editing. To do so, we propose EdiBERT, a bidirectional transformer that re-samples image patches conditionally to a given image. Using one generic objective, we show that the model resulting from a single training matches state-of-the-art GANs inversion on several tasks: image denoising, image completion, and image composition. We also provide several insights on the latent space of vector-quantized auto-encoders, such as locality and reconstruction capacities. The code is available at https://github.com/EdiBERT4ImageManipulation/EdiBERT.

## 1 Introduction

Significant progress in image generation has been made in the past few years, thanks notably to Generative Adversarial Networks (GANs) (Goodfellow et al., 2014). For example, the StyleGAN architecture (Karras et al., 2019; 2020b) yields state-of-the-art results in data-driven unconditional generative image modeling. Empirical studies have also shown the usefulness of GANs' architecture when it comes to image manipulation. By following specific directions in the latent space, one can modify an image attribute such as gender, age, the pose of a person (Shen et al., 2020), or the angle (Jahanian et al., 2019). However, since the whole picture is generated from a Gaussian vector, changing some undesired elements while keeping the others frozen is difficult. To solve this problem, edition algorithms involving optimization procedures have been proposed (Abdal et al., 2019; 2020) but with one main caveat: the results are not convincing when manipulating complex visuals (Niemeyer & Geiger, 2021) (cf. experimental section for visual results).

Independently, Van den Oord et al. (2017) propose VQVAE, a promising latent representation by training an encoder/decoder using a discrete latent space. The authors demonstrate the possibility to embed images in sequences of discrete tokens borrowing ideas from vector quantization (VQ), paving the way for the generation of images with autoregressive transformer models (Ramesh et al., 2021; Esser et al., 2021b). Building on this litterature, we argue that one of the benefits of this representation is that each token in the sequence is mostly coding for a localized patch of pixels (see section 3.5), thus opening the possibility for an efficient localized latent edition.

| Denoising | Completion | Compositing | Scribble-edit | Crossover |

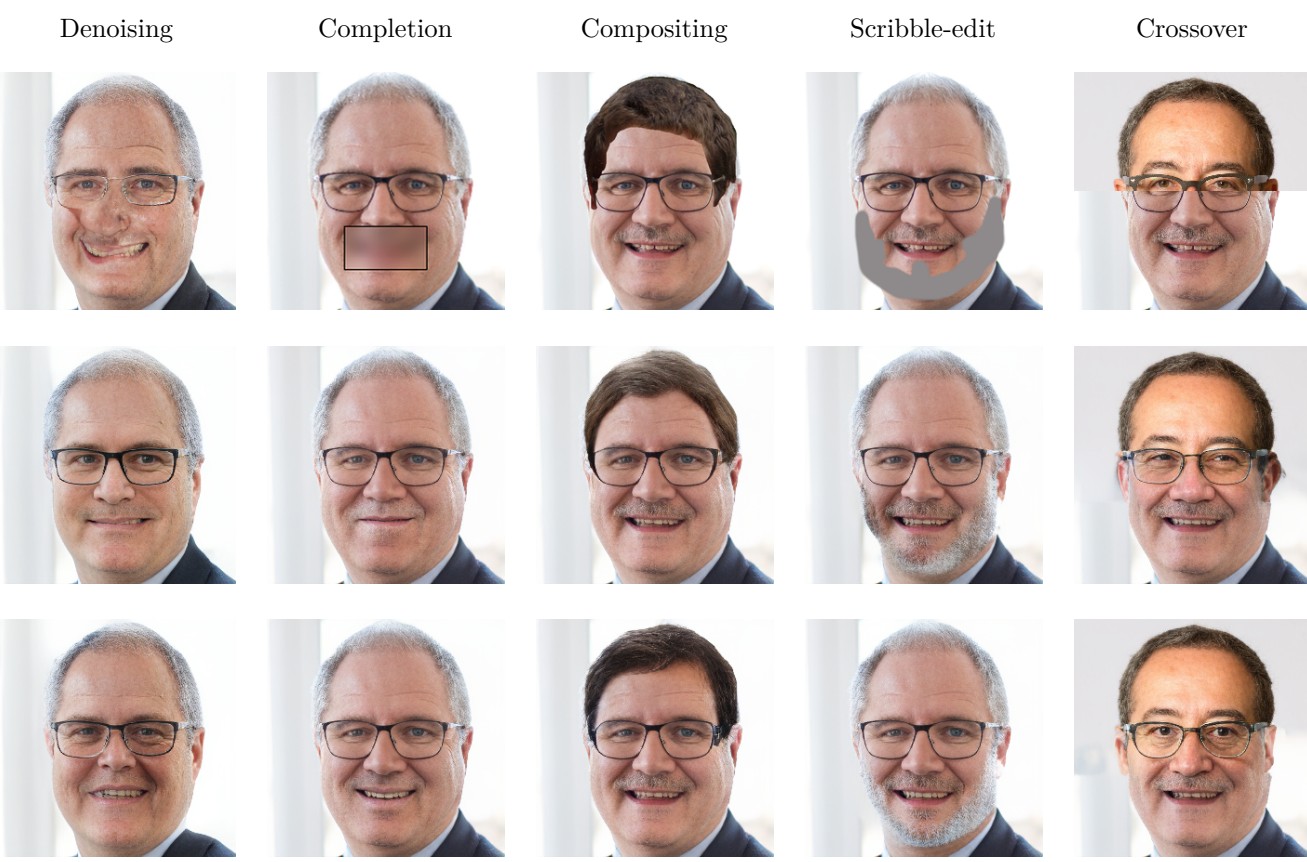

Figure 1: Using a single and straightforward training, EdiBERT can tackle a wide variety of different tasks in image editing. In this image, the top row is the input, while the second and third rows are different samples from EdiBERT, showing realism, consistency, and variety.

Aiming to build a unified approach for image manipulation, we propose a method that leverages both the spatial property of the discrete vector-quantized representation and the use of model that performs attention on the whole image. To do so, we train a bi-directionnal transformer network based on ideas from the language model BERT (Devlin et al., 2019), naming EdiBERT the resulting model. During training, EdiBERT tries to recover the original tokens of a perturbed sequence through a bidirectional attention schema. In computer vision, this approach has mainly been studied in the context of self-supervised representation learning (Bao et al., 2022; He et al., 2022). We advocate that training a single model using this generic objective provides a sounded way to obtain a model able to tackle several editing tasks. Finally, to practically handle these tasks, we also derived two different sampling algorithms: one dedicated for image denoising and editing, and a second one for inpainting.

To better visualize the abilities of EdiBERT after a single training, we show in Figure 1 that the same model can now be used in many different image manipulation tasks such as denoising, inpainting (or completion), compositing, and scribble-based editing.

To sum up, our contributions are the following:

+ We analyze the VQ latent representations and illustrate their spatial properties, and show how to improve the reconstruction capabilities of VQGAN, using a post-processing procedure that better recovers the pixel content outside of the edited region.

+ We show how to derive two different sampling algorithms from a single bidirectional transformers: one for the task of image denoising where the locations of the edits are unknown, and a second one for inpainting or image compositing where the mask specifying the area to edit is known.

+ Finally, we show that using this generic simple training algorithm along with its companion post-processing allow us to achieve competitive results on various image manipulation tasks.

## 2 Related work

We start this section by introducing transformer models for image generation. Then, we motivate the use of the VQ representation and bidirectional models for image manipulation.

### 2.1 Autoregressive image generation

The use of autoregressive transformers in the field of generative modeling (Parmar et al., 2018) has been made possible by two simultaneous research branches. First, the extensive deployment of attention mechanisms such as non-local means algorithms (Buades et al., 2005), non-local neural networks (Wang et al., 2018), and also attention layers in GANs (Zhang et al., 2019; Hudson & Zitnick, 2021). Second, the development of both classifiers and generative models sequentially inferring pixels via autoregressive convolutional networks such as PixelCNN (Van den Oord et al., 2016a;b). The self-attention mechanism (Vaswani et al., 2017), which now become ubiquitous in computer vision, is quickly recalled here: a sequence $X \in \mathbb{R}^{L \times d}$, where L is length of the sequence, is mapped by a position-wise linear layer to a query $Q \in \mathbb{R}^{L \times d_k}$, a key $K \in \mathbb{R}^{L \times d_k}$ and a value $V \in \mathbb{R}^{L \times d_v}$. The self-attention layer is then:

$$\text{attn}(Q, K, V) = \text{softmax}(\frac{QK^t}{\sqrt{d_k}})V \in \mathbb{R}^{L \times d_v} \tag{1}$$

If autoregressive transformers allow a principled log-likelihood estimation of the data, attention layers have a complexity scaling with the square of the sequence length, a clear bottleneck to scale to high-resolution images. To reduce the size of these sequences, Van den Oord et al. (2017) proposed the use of discrete representation. In this framework, an encoder $E$, a decoder $D$, and a codebook/dictionary $Z$ are learned simultaneously to represent images with a single sequence of tokens. Esser et al. (2021b) later trained an autoregressive model on these token sequences, stressing that high-capacity transformers can generate realistic high-resolution images. The framework consists of three steps:

1. Training simultaneously a set of encoder/decoder/codebook $(E, D, Z)$, by combining reconstruction, commitment and adversarial losses. The reconstruction loss is a perceptual distance (Zhang et al., 2018). The commitment loss (Van den Oord et al., 2017) pushes the codebook towards the output of the encoder using a quantization loss. The adversarial loss is the Vanilla GANs loss defined in (Goodfellow et al., 2014). The training objective becomes :

$$E^\star, D^\star, Z^\star = \underset{E,D,Z}{\arg\min} \left[ L_{\text{rec.}}(E, D, Z) + L_{\text{commit.}}(E, Z) + \lambda L_{\text{adv.}}(\{E, D, Z\}) \right]. \tag{2}$$

2. Training an autoregressive transformer to maximize the log-likelihood of the encoded sequences.

3. At inference, sampling a sequence with the transformer and decoding it with the decoder $D$.

This vector-quantized representation was later improved by Yu et al. (2021a) and used by Yu et al. (2022) to create PARTI, a state-of-the-art text-to-image generative model. Interestingly, our work EdiBERT builds on top of the first step of VQGAN, and also requires the training of the triplet $(E, D, Z)$ following equation 2.

### 2.2 Bidirectional attention

The main property of autoregressive models is that they only perform attention on previous tokens, making them inadequate when dealing with image manipulation (Esser et al., 2021a). Some works alleviate this bias

in different ways. Yang et al. (2019) learn an autoregressive model on random permutations of the ordering. Cao et al. (2021) propose a model where missing tokens are inferred autoregressively, conditionally to the set of kept tokens. Similarly, Wan et al. (2021) use an auto-regressive procedure conditioned on the masked image, while Yu et al. (2021b) use BERT training with [MASK] tokens and Gibbs sampling. If this setting is ideal for tasks with masked tokens such as inpainting, it makes it ill-posed for scribble-editing and insertion without existing paired datasets. On the opposite, our EdiBERT tackles all tasks without any need for supervision. Finally, Esser et al. (2021a) train ImageBART, a multinomial diffusion process (Hoogeboom et al., 2021) in the discrete latent space of VQGAN. Each generated sequence is conditioned on the previous one and performs attention to the whole image. However, this method is computationally heavy since it requires making $N \times L$ inferences, where $N$ is the number of generated sequences and $L$ is the number of tokens in the sequence. A more efficient way to perform bidirectional attention for image generation has been proposed in MaskGIT (Chang et al., 2022). MaskGIT consists of training with a BERT-like objective (Devlin et al., 2019) on sequences randomly perturbed with [MASK] tokens, and generating images with a parallel decoding scheme. Similarly, Zhang et al. (2021) propose to use a masking-based strategy to perform conditional image editing with bidirectional attention mechanisms. However, they still require specific conditional data to learn their model editing model. In this paper, we argue that by performing bidirectional attention over all the tokens and learning with a denoising objective (tokens perturbed by randomization, not with [MASK] tokens), it is possible to train a single model tackling many editing tasks.

## 2.3 Unifying image manipulation

Initially, image manipulation methods were implemented without any trainable parameters. Image completion was tackled using nearest-neighbor techniques along with a large dataset of scenes (Hays & Efros, 2007). As to image insertion, blending methods were widely used, such as the Laplacian pyramids (Burt & Adelson, 1987). In recent years, image manipulation has benefited from the advances of deep generative models. A first line of work has consisted of gathering datasets of corrupted and target images to train conditional generative models. By doing so, one can therefore learn a mapping from any corrupted image to a real one. For example, Liu et al. (2021) proposes an encoder-decoder architecture for sketch-guided image inpainting. However, in all cases, a dataset with both types of images is required, therefore limiting the applicability.

To avoid this dependency, a second idea - known as GAN inversion methods - leverages pre-trained unconditional GANs. They work by projecting edited images on the manifold of real images learned by the pre-trained GAN. It can be solved either by optimization (Abdal et al., 2019; 2020; Daras et al., 2021), or with an encoder mapping to the latent space (Chai et al., 2021; Richardson et al., 2021; Tov et al., 2021). Pros of these GAN-based methods are that one benefits from the outstanding properties of StyleGan, state-of-the-art in image generation. However, these methods rely on a task-specific loss function that needs to be defined and optimized. More recently, another line of research is based on the development of score-based models (Song et al., 2020): Meng et al. (2022) use Langevin's dynamics for image edition and, (Esser et al., 2021a) combine discrete diffusion models (Hoogeboom et al., 2021; Austin et al., 2021) with the discrete vector-quantized representations from VQGANs.

# 3 Motivating EdiBERT for image editing

This section gives a global description of the proposed EdiBERT model. We start with notations before describing the different steps leading to the BERT-based edition.

## 3.1 Discrete auto-encoder VQGAN

Let $I$ be an image with width $w$, a height $h$, and a number $c$ of channels. $I$ thus belongs to $\mathbb{R}^{h \times w \times c}$. Let $(E, D, Z)$ be respectively the encoder, decoder, and codebook defined in VQVAE and VQGAN (Van den Oord et al., 2017; Esser et al., 2021b). The codebook $Z$ consists of a finite number of tokens with fixed vectors in an embedding space: $Z = \{t_1, \ldots, t_N\}$ with $t_k \in \mathbb{R}^d$ and $N$ being the cardinality of the codebook.

For any given image $I$, the encoder $E$ outputs a vector $E(I) \in \mathbb{R}^{L \times d}$, which is then quantized and reshaped into a sequence $s$ of length $L$ as follows:

$$s = (\arg\min_{z \in Z} \|E(I)_1 - z\|, \ldots, \arg\min_{z \in Z} \|E(I)_L - z\|) = Q_Z(E(I)), \qquad (3)$$

where $E(I)_l = E(I)_{l,:} \in \mathbb{R}^d$ is the feature vector of $E(I)$ at position $l$, and $Q_Z$ refers to the quantization operation using the codebook $Z$. Recall that, after the quantization step, one gets a sequence composed of $L$ codebook elements, thus $s \in Z^L$. After we feed the codebook embeddings to the decoder $D$, the reconstructed image becomes $\hat{I} = D(Q_Z(E(I)))$.

Let's note $\mathcal{D}$, the available image dataset. From a pre-trained encoder $E$ and codebook $Z$, one can transform the image dataset $\mathcal{D}$ into a dataset of token-sequences $\mathcal{D}_S := \{Q_Z(E(I)), I \in \mathcal{D}\}$. When learning transformers on sequences of tokens, the practitioner is directly working with $\mathcal{D}_S$.

## 3.2 Learning sequences with autoregressive models

The following sections aim at motivating the training objective for the EdiBERT model. To begin with, let $p_\theta$ be a transformer model parameterized with $\Theta$ trained on $\mathcal{D}_S$. For each position $i$ in $s$, we note $p_\theta^i(.|s)$, the modeled distribution of tokens conditionally to $s$.

When training an autoregressive transformer on the discrete sequences of tokens $\mathcal{D}_S$ (Esser et al., 2021b), one needs to compute the likelihood $p_\theta(s)$ of each given sequence $s = (s_1, \ldots, s_L) \in \mathcal{D}_S$ as follows:

$$p_\theta(s) = \prod_{i=1}^{L} p_\theta^i(s_i | s_{<i}), \quad \text{with } s_{<i} = (s_1, \ldots, s_{i-1}). \qquad (4)$$

Conditional distributions $p_\theta^i(s_i | s_{<i})$ are computed using a causal left-to-right attention mask. The final objective of the autoregressive model is to find the best set of parameters within $\Theta$:

$$\arg\max_{\theta \in \Theta} \mathbb{E}_{s \in \mathcal{D}_s} \log p_\theta(s). \qquad (5)$$

**Limitations of the model.** If this setting is well suited for unconditional image generation, it is ill-posed for image manipulation tasks, as shown by Esser et al. (2021a). In the case of scribble-based editing, or inpainting, one wants to resample tokens conditionally to the whole image, so that the model has all the information at its disposal.

## 3.3 A unique training objective for EdiBERT.

Let us define the training objective for EdiBERT. For a sequence $s = (s_1, ..., s_L)$, a function $\varphi$ randomly selects a subset of $k$ indices $\{\varphi_1, ..., \varphi_k\}$ where $\varphi_k < L$. At each selected position $\varphi_i$, a perturbation is applied on the token $s_{\varphi_i}$. We attribute a random token with probability $p$, or keep the same token with probability $1 - p$. Consequently, the perturbed token $\tilde{s}_{\varphi_i}$ becomes:

$$\tilde{s}_{\varphi_i} = \mathbb{U}(Z) \quad \text{with probability } p,$$
$$\tilde{s}_{\varphi_i} = s_{\varphi_i} \quad \text{with probability } 1 - p,$$

where $\mathbb{U}(Z)$ refers to the uniform distribution on the space of tokens $Z$. Similarly to Bao et al. (2022), the sampling function $\varphi$ is defined with a 2D selection strategy, and the positions are selected by drawing random 2D rectangles, see in Figure 2. Contrarily to Bao et al. (2022) and Devlin et al. (2019), we only use random tokens from the codebook but no [MASK] tokens. We argue this setting corresponds more to the cases of denoising and editing, where tokens have to be sampled conditionally to an entire perturbed sequence.

Let us now call $\tilde{s} = (s_1, \ldots, \tilde{s}_{\varphi_1}, \ldots, \tilde{s}_{\varphi_k}, \ldots, s_L)$ the perturbed sequence, and $\tilde{\mathcal{D}}_s = \{\tilde{s}, \ s \in \mathcal{D}\}$ the perturbed dataset. The training of EdiBERT optimizes the following objective :

$$\arg\max_{\theta \in \Theta} \mathbb{E}_{\tilde{s} \in \tilde{\mathcal{D}}_s} \frac{1}{k} \sum_{i=1}^{k} \log p_\theta^i(s_{\varphi_i} | \tilde{s}). \qquad (6)$$

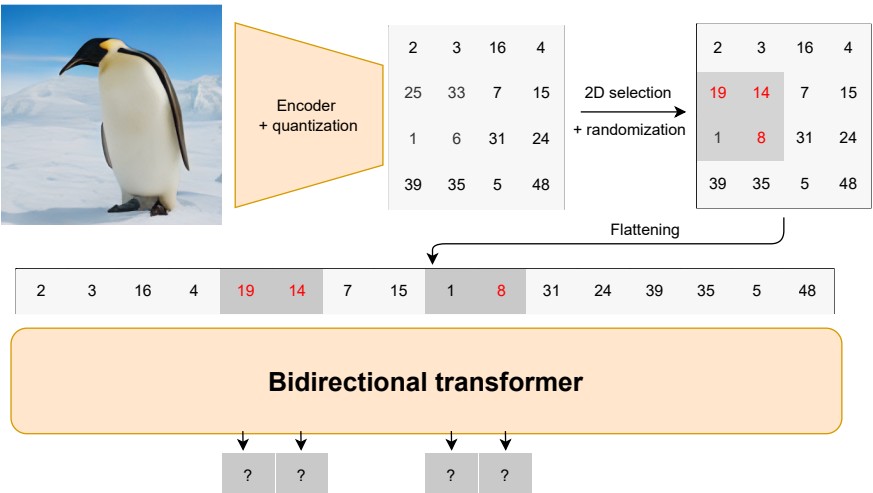

Figure 2: The 2D selection and randomization strategy for the training of our bidirectional transformer: EdiBERT is trained on sequences where localized patch of tokens have been perturbed.

Contrary to equation 5, we note that the objective in equation 6 does not require a causal left-to-right attention. Instead, the attention can be performed over the whole input sequence.

**Sampling from EdiBERT:** Wang & Cho (2019) show that it is possible to generate realistic samples with a BERT model starting with random initialization. However, compared with standard autoregressive language models, the authors stress that BERT generations are more diverse but of slightly worse quality. Building on the findings of Wang & Cho (2019), we do not aim to use BERT for pure unconditional sequence generation but rather improve an already existing sequence of tokens. In our defined EdiBERT model, for any given position $i \in s$, a token will be sampled according to the multinomial distribution $p_\theta^i(.|s)$.

### 3.4    On the locality of Vector Quantization encoding

In this paper, we argue that one of the main advantages of EdiBERT comes from the VQ latent space proposed by Van den Oord et al. (2017) where each image is encoded in a discrete sequence of tokens. In this section, we illustrate with simple visualizations the property of this VQGAN encoding. We explore the spatial correspondence between the position of the token in the sequence and a set of pixels for the encoded image. We aim at answering the following question: do local modifications of the image lead to local modifications of the latent representation and *vice versa*?

**Modifying the image.**    To answer this question, images are voluntarily perturbed with grey masks ($i \longrightarrow i_m$). Then, we encode the two images, quantize their representation using a pre-trained codebook, and plot the distance between the two latent representations: $\|Q_Z(E(i)) - Q_Z(E(i_m))\|_2^2$. The results are shown in the first row in Figure 3. Due to the large receptive field of the encoder, tokens can be influenced by distant parts of the image: the down-sampled mask does not recover all of the modified tokens. However, tokens that are largely modified are either inside, or very close to the down-sampled mask.

**Modifying the latent space.**    To understand the correspondence between tokens and pixels, we stress how one can easily manipulate images using the discrete latent space. In Figure 3, we show that cutting a specific area of a source image to insert it in a different location of another image is possible only by replacing the corresponding tokens in both sequences. This spatial correspondence between VQGANs' latent space and the image is interesting for localized image editing tasks, *i.e.* tasks that require modifying only a subset of pixels without altering the other ones.

**Modifying the latent space via the image**

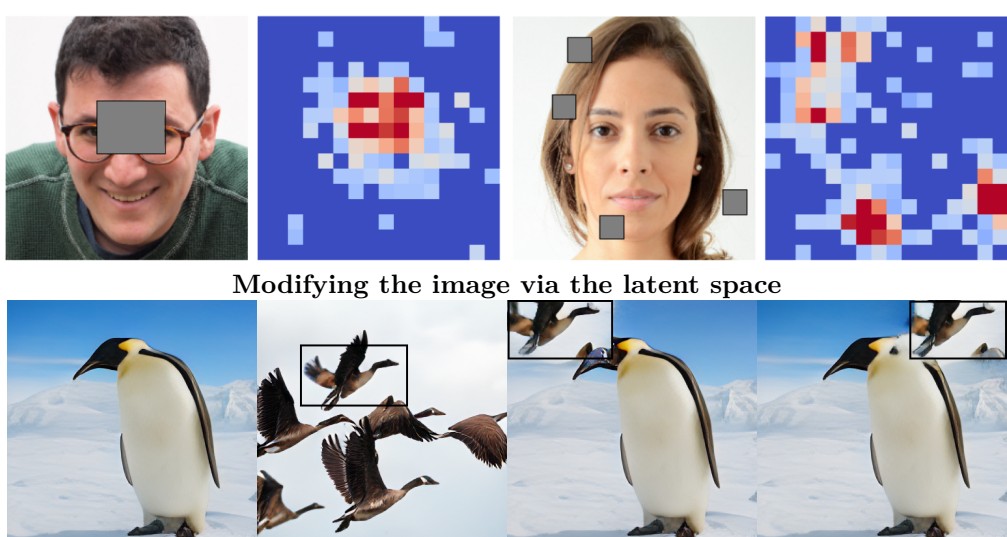

**Modifying the image via the latent space**

Figure 3: Each token in the sequence is tied to a small spatial area in the decoded image. In the 1st row: we voluntarily perturb images and display the variations among the tokens in the latent space. The heatmaps represent the distance (red is high) between the tokens of the original image and the tokens of the perturbed image. In the 2nd row: we stress how collages of images can easily be done with this discrete latent representation: third and fourth images are generated by the decoder from a latent space collage.

### 3.5 On the reconstruction capabilities of Vector Quantization encoding

A limit of the framework resides in the use of the vector quantization operation and the induced loss of information. Indeed, we observe in Figure 4 that VQGAN struggles to reconstruct high-frequency details, for example complex backgrounds on FFHQ dataset (Karras et al., 2019). To improve the reconstruction capabilities of VQGANs, we propose a simple optimization procedure over the latent space vectors.

The objective is to find the latent vectors that minimize the LPIPS (Zhang et al., 2018) between the target image and the decoded reconstruction. We initialize the procedure from the output of the encoder $E(I)$, and optimize the objective with gradient descent. Figure 4 shows how this procedure improves the inversion capabilities of VQGAN to make it better than GAN inversion methods (Abdal et al., 2020). A potential explanation of the limited reconstruction capabilities of VQGAN is displayed in Figure 5: the latent vectors of the codebook might suffer from a very low rank. The optimization procedure seems to solve this since the latent vectors span much more dimensions of the embedding space after a few hundred optimization steps.

## 4 Image editing with EdiBERT

**Baselines.** For each task, we run comparisons with baselines and state-of-the-art models based on GANs inversion methods. On FFHQ, we compare to ImageStyleGAN2++ (Abdal et al., 2020) on pre-trained StyleGANs: StyleGAN2 (Karras et al., 2020b) and StyleGAN2-ADA (Karras et al., 2020a). Besides, we run the solution proposed by Chai et al. (2021) where a StyleGAN2 model is inverted using a trained encoder. Finally, we use In-Domain GAN (Zhu et al., 2020), a hybrid method combining an encoder with an optimization procedure minimizing reconstruction losses. We also compare to Co-Modulated GANs (Zhao et al., 2020), a conditional GAN for inpainting.

**Metrics.** We follow the work of Chai et al. (2021) and use metrics assessing both fidelity and distribution fitting. The masked L1 metric (Chai et al., 2021) measures the closeness between the generated image and the source image outside the edited areas. The density/coverage metrics (Naeem et al., 2020) are robust

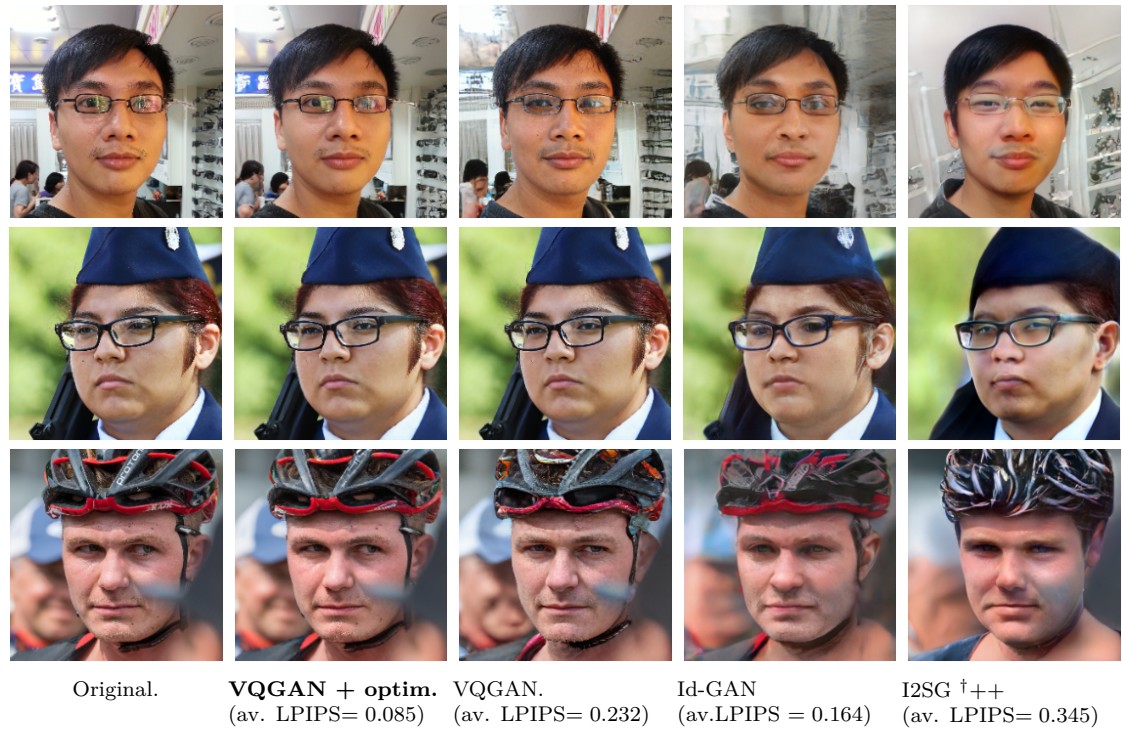

| Original. | **VQGAN + optim.** (av. LPIPS= 0.085) | VQGAN. (av. LPIPS= 0.232) | Id-GAN (av.LPIPS = 0.164) | I2SG †++ (av. LPIPS= 0.345) |

Figure 4: Comparison of reconstruction capabilities of VQGAN + optimization to two GANs inversion methods such as Id-GAN (Zhu et al., 2020) and I2SG†++ (Abdal et al., 2020). Averaged LPIPS are computed on the validation set FFHQ.

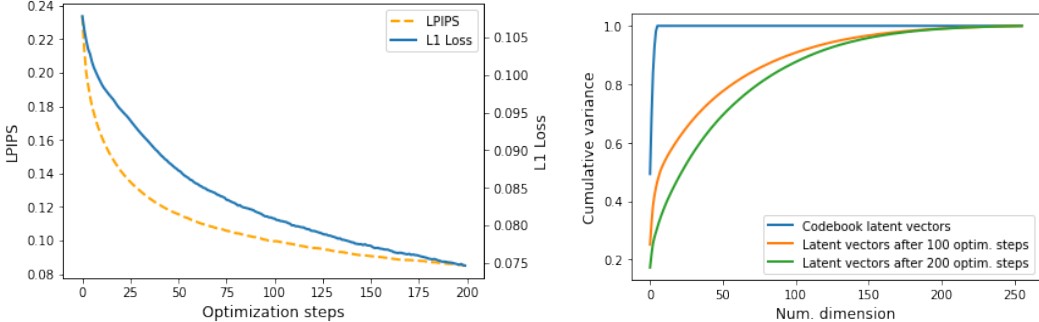

Figure 5: Analysis of reconstruction capabilities of VQGAN. On the left, we see that both the L1 and perceptual loss (LPIPS) between original and reconstructed images significantly decrease when optimizing LPIPS over the latent vectors of VQGAN. This may be a consequence of a higher number of dimensions spanned by the latent vectors (on the right), after the optimization (allowing for more complex reconstructions).

versions of precision/recall metrics. Intuitively, density measures fidelity while coverage measures diversity. Finally, the FID (Heusel et al., 2017) quantifies the distance between generated and target distributions. Moreover, we perform a user study on FFHQ image compositing. More details and quantitative results on LSUN Bedroom are presented in Appendix.

## 4.1 Localized image denoising

Image denoising aims to improve the quality of a pre-generated image or improve a locally perturbed one. The model has to work without information on the localization of the perturbations. This means we need

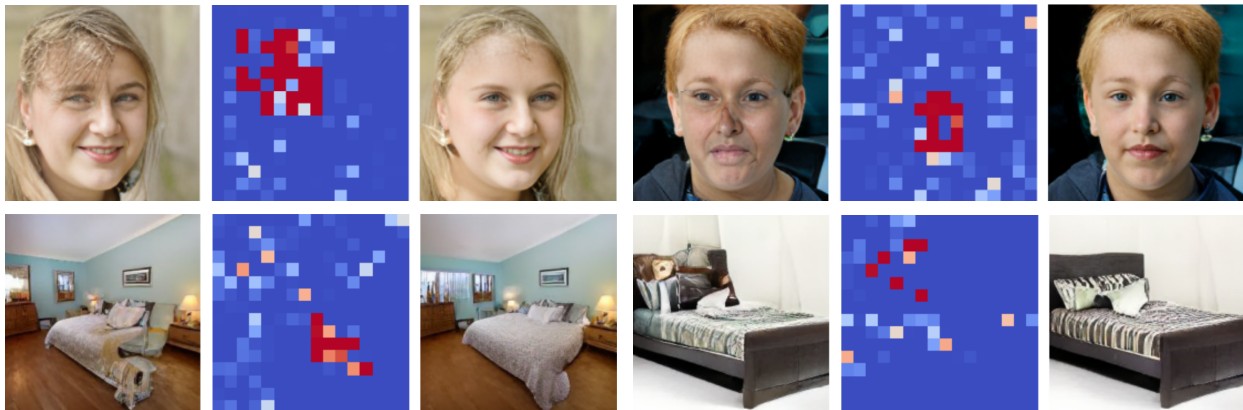

Figure 6: Image denoising with EdiBERT: the color in the 4 different heatmaps is proportional to the negative likelihood of the token. Tokens with a lower likelihood appear in red in the heatmap and have a higher probability of being sampled and edited. Consequently, conditional distributions output by EdiBERT are an efficient tool to detect anomalies and artifacts in the image.

to find and replace the perturbed tokens with more likely ones to recover a realistic image. Thus, given a sequence $s = (s_1, \ldots, s_L)$, we want to:

1. Detect the tokens that do not fit properly in the sequence $s$.

2. Change them for new tokens increasing the likelihood of the new sequence.

We desire a significantly more likely sequence with as few as possible token amendments. To do so, we measure the likelihood of each token $s_i$ based on the whole sequence $s$, aiming to compute $p(s_i|s)$, and replace the least-probable tokens considering them independently. That is, we propose to use the conditional probability output by the model in order to detect and sample the *less likely* odd tokens. Some examples of image denoising are presented in Figure 6, and we observe that EdiBERT is able to detect artifacts and replace them with more likely tokens. The full algorithm is given in the Algorithm 1.

---

**Algorithm 1:** Image denoising

**Requires:** Sequence $(s_1, \ldots, s_L)$, BERT model $p_\theta$, number of iterations $T$;
**for** *iterations in [0,T]* **do**
    Compute $p_i = \text{logit}(-p_\theta^i(s_i|s)), \forall i \in [1, L]$ ;
    Sample $p \sim (p_1, \ldots, p_l)$ (*less likely position*);
    Sample $t \in Z \sim p_\theta^p(\cdot|s)$ ;
    Insert sampled token: $s_i \leftarrow t$ ;
**end**
Image $\leftarrow$ Decoder($s$);
**Result:** Image

---

## 4.2 Image inpainting

In this setting, we have access to a masked image $i_m \in \mathbb{R}^{h \times w \times c}$ along with the location of the binary mask $m \in \mathbb{R}^{h \times w}$. $i_m$ has been obtained by masking an image $i \in \mathbb{R}^{h \times w \times c}$ as follows: $i_m = i \odot m$ with $\odot$ pointwise multiplication. The goal of image inpainting is to generate an image $\hat{i}$ that is both realistic (high density) and conserves non-masked parts, that is $\hat{i} \odot (1 - m) = i \odot (1 - m)$.

Among the different tasks in image manipulation, image inpainting stands out. Indeed, when masking a specific area of an image, one shall not consider the pixels within the mask to recover the target image.

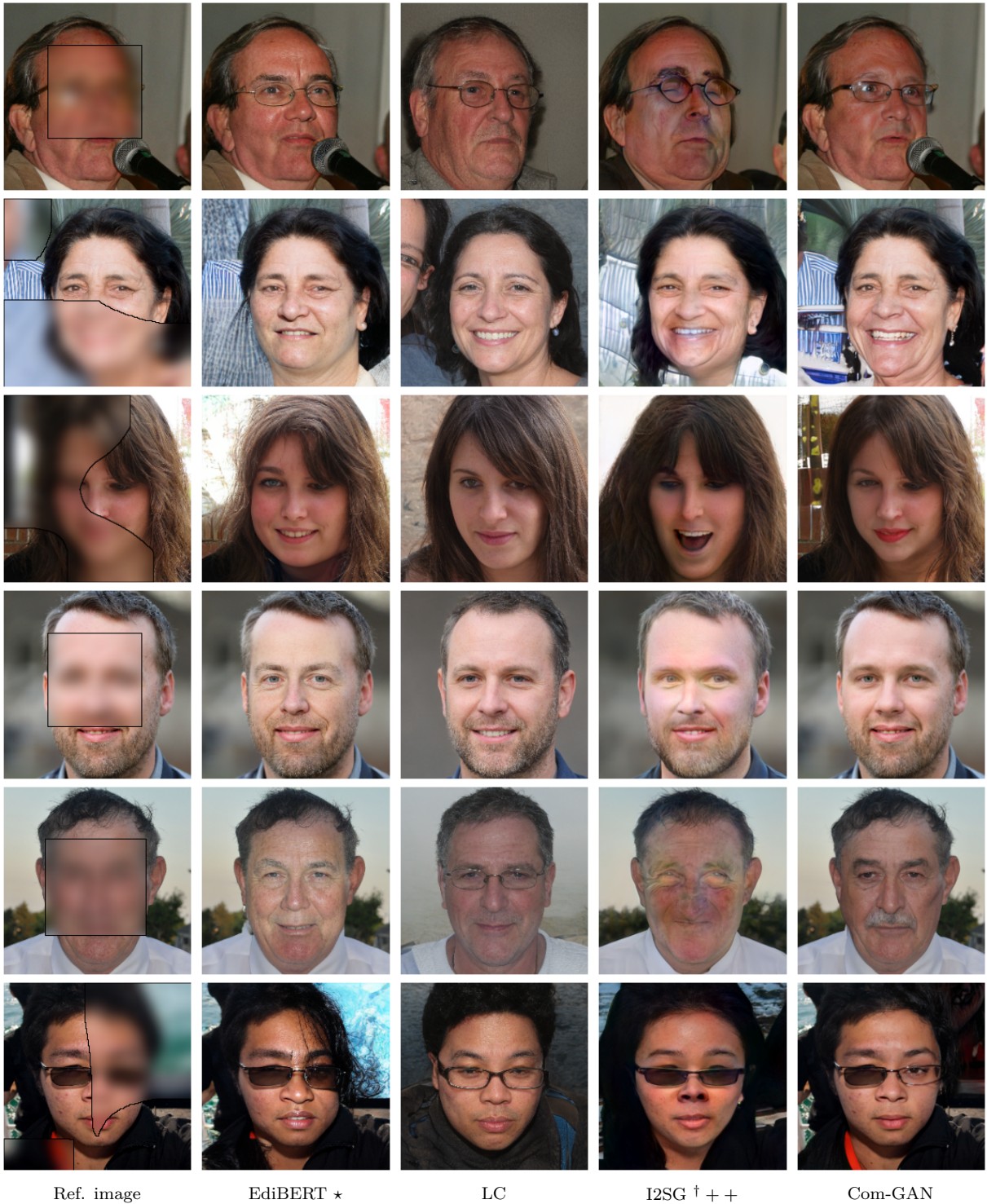

| Ref. image | EdiBERT ⋆ | LC | I2SG † + + | Com-GAN |

Figure 7: Image inpainting comparisons on FFHQ. EdiBERT performs better than inversion methods such as LC (Chai et al., 2021) and I2SG (Abdal et al., 2020). Note that Com-GAN (Zhu et al., 2020) is specialized for image inpainting and was trained on pair datasets (masked image, target image), it can not perform other image editing tasks.

The image inpainting task thus requires specific care to reach a state-of-the-art performance; this is why we added five different elements to our approach, and validated these elements with visual results in Figure 13.

1. **Randomization:** to erase the mask influence, the tokens within the mask are given random values.

2. **Dilation of the mask:** as shown in Figure 3, some tokens outside of the down-sampled mask in the latent space are also impacted by the mask on the image. Modifying only tokens inside the down-sampled mask might not be enough and could lead to images with irregularities on the borders. As a solution, we apply a dilation on the down-sampled mask and show in Figure 13 that it helps better blend the target image's completion since the boundaries are removed.

3. **Spiral ordering:** since there is no pre-defined ordering of positions in EdiBERT, one can look for an optimal sampling of positions. We argue that by sampling positions randomly, one does not fully leverage the spatial location of the mask. Instead, we propose a spiral ordering that goes from the border to the inside of the mask. Qualitative and quantitative results in Figure 13 and Table 2 confirm the advantage of this ordering.

4. **Periodic image collage:** to preserve fidelity to the original image, we periodically perform a collage between the masked image and the decoded image. We observed in Figure 13, that without this collage trick, the reconstruction can diverge too much from the input image.

5. **Online optimization on latent sequences:** to improve fidelity to the masked image $i_m$, the final stage of the algorithm consists in an optimization procedure on the latent sequence $s \in \mathbb{R}^{h \times w \times d}$. The objective function is defined as:

$$L = L_p\big((D(s) - i_m) \odot m\big) + L_p\big((D(s) - D(s^0)) \odot (1 - m)\big) \tag{7}$$

where $L_p$ is a perceptual loss (Zhang et al., 2018), and $s^0$ is the initial sequence from EdiBERT. Intuitively, the first term enforces the decoded image to get closer to the masked image $i_m$, while the second term is a regularization enforcing the decoded image to stay similar to the completion proposed by the transformer's likelihood. We illustrate in Figure 5 and Figure 13 that the optimization leads to a better-preserved source image.

**Analyzing the results:** we see from Table 1 and Figure 7 that the specialized method com-GAN (Zhao et al., 2020) outperforms non-specialized methods on image inpainting. This was expected since it is the only method that has been trained specifically for this task. Note that the trained model co-mod GAN cannot be used in any other image manipulation task. Compared with the non-specialized method, EdiBERT always provides better fidelity to the source image (lower Masked L1) and realism (best FID and top-2 density). An ablation study is available in Table 2 and validates our choices. Finally, more details regarding the sampling algorithm for the task of inpainting are given in Appendix.

### 4.3 Image composition

In this setting, we have access to a non-realistically edited image $i_e \in \mathbb{R}^{h \times w \times c}$. The edited image $i_e$ is obtained by a composition between a source image $i_s \in \mathbb{R}^{h \times w \times c}$ and a target image $i_t \in \mathbb{R}^{h \times w \times c}$. The target image can be a user-drawn scribble or another real image in the case of image compositing. Besides, pixels are edited inside a binary mask $m \in \mathbb{R}^{h \times w}$, which indicates the areas modified by the user. Thus, the final edited image is computed pointwise as:

$$i_e = i_s \odot m + i_t \odot (1 - m). \tag{8}$$

Image composition aims to transform an edited image $i_e$ to make it more realistic and faithful without limiting the changes outside the mask. We note the source image $i_s$ outside the mask and the edits of the target image $i_m$ for the inserted elements in the edition mask. Three tasks fall under this umbrella: *scribble-based editing*, *image compositing*, and *image crossovers*.

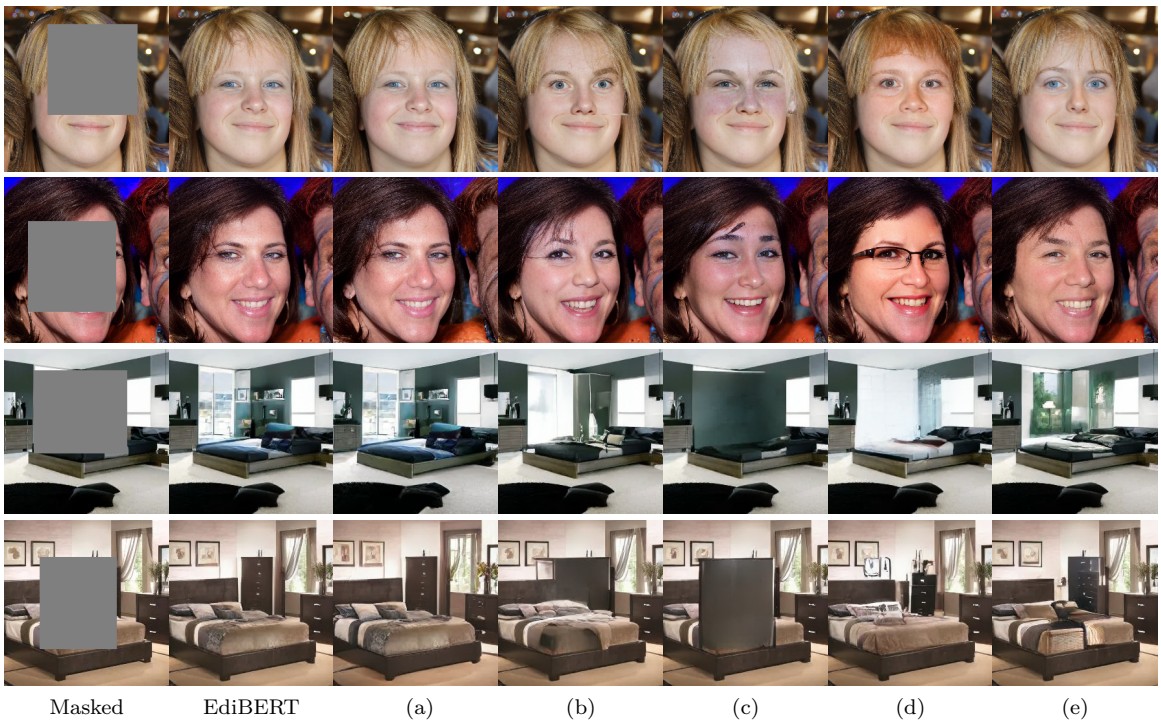

| Masked | EdiBERT | (a) | (b) | (c) | (d) | (e) |

Figure 8: Ablation study for inpainting. Components removed are (a) optimization, (b) dilation, (c) randomization, (d) collage, (e) spiraling (random order instead). Optimization improves fidelity to the source image, while the other components help increase image quality.

Table 1: Image inpainting and compositing on FFHQ $256 \times 256$. Com-GAN is a model specific for image inpainting, ID-GAN handles several editing tasks but not inpainting, while other methods handle both. We remove I2SG++ from the user study, since I2SG$^\dagger$++ is the same method with a better GAN backbone, *i.e.* StyleGAN2-ADA (Karras et al., 2020a). **Bold:** $1^{st}$ **rank**, blue: $2^{nd}$ rank.

|  | Inpainting | | | | Compositing | | |
|---|---|---|---|---|---|---|---|
|  | Masked L1 $\downarrow$ | FID $\downarrow$ | Dens. $\uparrow$ | Cover. $\uparrow$ | Masked L1 $\downarrow$ | Dens. $\uparrow$ | User study $\uparrow$ |
| I2SG++ (Abdal et al., 2020) | 0.0767 | 23.6 | 0.99 | 0.88 | 0.0851 | 0.77 | - |
| I2SG$^\dagger$++ Abdal et al. (2020) | 0.0763 | 22.1 | 1.25 | 0.91 | 0.0866 | **1.07** | 8.3% |
| LC (Chai et al., 2021) | 0.1027 | 27.9 | 1.12 | 0.84 | 0.1116 | 1.00 | 14.8% |
| EdiBERT ⋆ | 0.0290 | 13.8 | 1.16 | 0.98 | **0.0307** | 0.94 | **61.2**% |
| Com-GAN (Zhu et al., 2020) | **0.0086** | **10.3** | **1.42** | **1.00** | - | - | - |
| ID-GAN (Zhu et al., 2020) | - | - | - | - | 0.0570 | 0.75 | 15.7% |

Results of image compositing on FFHQ are presented in Table 1 and Figure 9. EdiBERT always has the lowest masked L1. We also present the results from a user study in Table 1. 30 users were shown 40 original and edited images, along with four results (EdiBERT and baselines). They were asked which one is preferable, accounting for both fidelity and realism. The survey shows that on average, users prefer EdiBERT over competing approaches. We give more visual results along with the detailed answers of the user study in Appendix.

Table 2: Inpainting: Ablation study on the components of EdiBERT sampling algorithm. EdiBERT (1st row) shows the best tradeoff between fidelity (masked L1) and quality (FID, density/coverage). **Bold:** $1^{st}$ **rank**, blue: $2^{nd}$ rank.

| Ordering | Optim-ization | Random-ization | Collage | Dilation | Masked L1 ↓ | FID ↓ | Density ↑ | Coverage ↑ |
|---|---|---|---|---|---|---|---|---|
| Spiral | ✓ | ✓ | ✓ | ✓ | 0.0201 | **19.4** | 1.14 | **0.96** |
| Random | ✓ | ✓ | ✓ | ✓ | 0.0206 | 20.7 | 1.13 | 0.95 |
| Spiral | X | ✓ | ✓ | ✓ | 0.0299 | 20.3 | 1.20 | 0.94 |
| Spiral | ✓ | X | ✓ | ✓ | 0.0198 | 20.5 | **1.26** | 0.92 |
| Spiral | ✓ | ✓ | X | ✓ | 0.0252 | 19.9 | 1.11 | 0.95 |
| Spiral | ✓ | ✓ | ✓ | X | **0.0197** | 23.3 | 0.96 | 0.91 |

## 5 Discussions

EdiBERT is a bidirectional transformers model that can tackle multiple editing tasks from one single training. One of the key elements of the proposed method is that it does not require having access to paired datasets (source, target), or unpaired image datasets. This property shows how flexible EdiBERT is and why it can be easily applied to different tasks. Overall, the proposed framework is simple and tractable: 1) train a VQGAN Esser et al. (2021b), 2) train an EdiBERT model following the objective defined in equation 6.

Interestingly, for simple applications, one can directly train EdiBERT based on the representations output by the VQGAN pre-trained on ImageNet. However, for more complex data or when dealing with multiple domains, one might have to train a specialized codebook, which requires a large auto-encoder and a lot of data. Another EdiBERT's drawback is related to the core interest of image editing. Since the tokens are predominantly localized, EdiBERT is perfectly suited for small manipulations that only require amending a few numbers of tokens. However, some manipulations such as zooms or rotations require changing large areas of the source image. In these cases, modifying a large number of tokens might be more demanding.

**Broader Impact Statement**

Similarly to other image generative models, EdiBERT might be used to create and propagate fake beliefs via deepfakes, as discussed in Fallis (2020).

## 6 Conclusion

In this paper, we demonstrated the possibility to perform several editing tasks by using the same pre-trained model. The proposed framework is simple and aims at making a step towards a unified model able to do any conceivable manipulation task on images. An exciting direction of research would be to extend the editing capabilities of EdiBERT to global transformations (*e.g.* zoom, rotation).

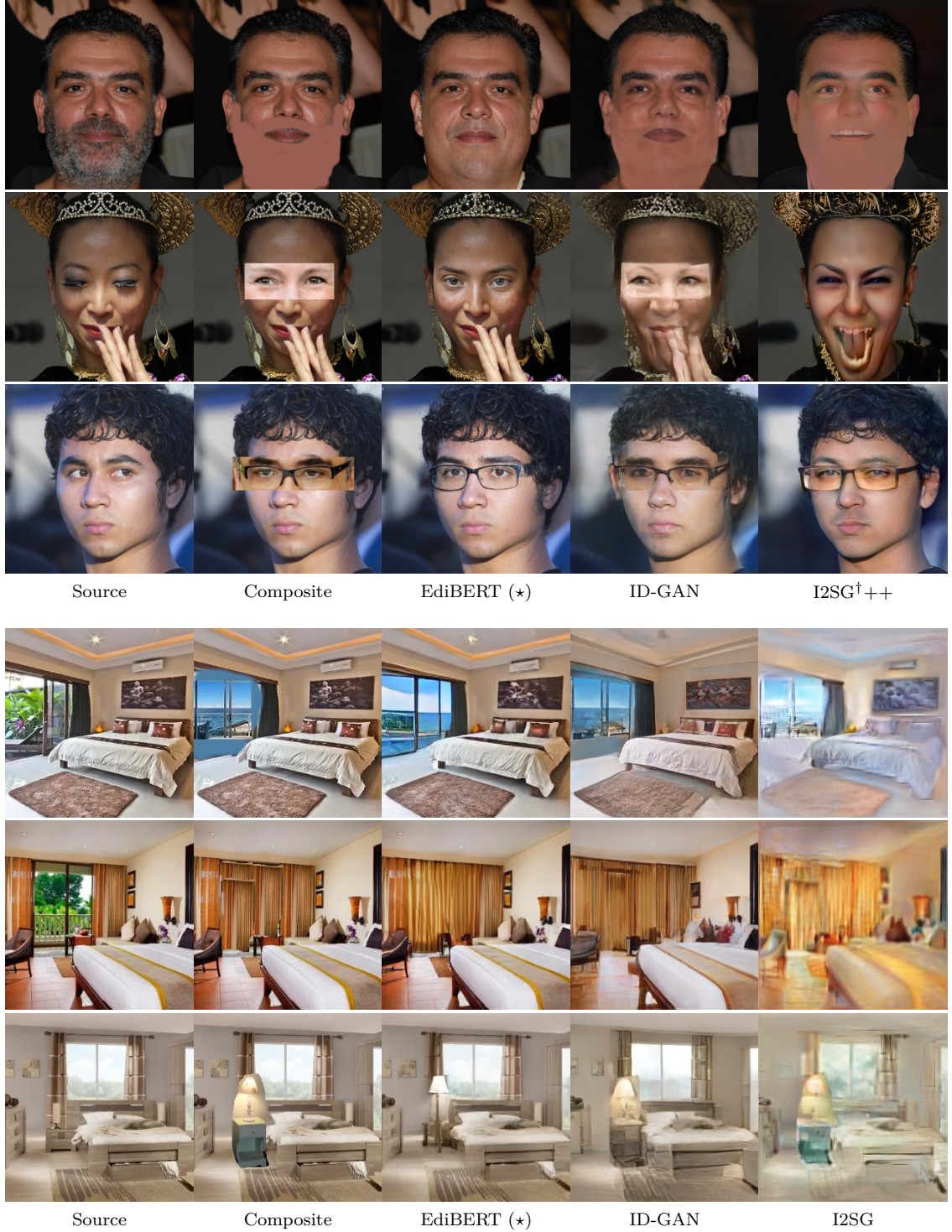

|  |  |  |  |  |
|---|---|---|---|---|
| Source | Composite | EdiBERT ($\star$) | ID-GAN | I2SG$^{\dagger}$++ |

| Source | Composite | EdiBERT ($\star$) | ID-GAN | I2SG |
|---|---|---|---|---|

Figure 9: Scribble-based editing and image compositing: comparison with ID-GAN (Zhu et al., 2020) and I2SG (Abdal et al., 2019). EdiBERT preserves better the fidelity to the source image while being also able to fit the inserted object properly. This confirms the quantitative results in Table 1, EdiBERT seems to be leading in both fidelity and realism.

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

# Appendix

## A   Implementation details

The code for the implementation of EdiBERT is available on GitHub at the following link https://github.com/EdiBERT4ImageManipulation/EdiBERT.

A pre-trained model on FFHQ is available on a linked Google Drive. Notebooks to showcase the model have also been developped.

### A.1   Training hyper-parameters

We use the same architecture than Esser et al. (2021b) for both VQGAN and transformer. On LSUN Bedroom and FFHQ, we use a codebook size of 1024. For the transformer, we use a model with 32 layers of width 1024.

To train the transformer with 2D masking strategy, we generate random rectangles before flattening $Q(E(I))$. The height of rectangles is drawn uniformly from $[0.2 \times h, 0.5 \times h]$. Similarly, the width of rectangles is drawn uniformly from $[0.2 \times w, 0.5 \times w]$. In our experiments, since we work at resolution $256 \times 256$ and follow the downsampling factor of 4 from Esser et al. (2021b), we have $h = w = 256/16 = 16$.

Tokens outside the rectangle are used as input, to give context to the transformer, but not for back-propagation. Tokens inside the rectangle are used for back-propagation. $p_{\mathrm{rand}} = 90\%$ of tokens inside the mask are put to random tokens, while $p_{\mathrm{same}} = 1 - p_{\mathrm{rand}} = 10\%$ are given their initial value. Although we did not perform a large hyper-parameter study on this parameter, we feel it is an important one. The lower $p_{\mathrm{rand}}$, the more the learned distributions $p_\theta^i(.|s)$ will be biased towards the observed token $s_i$. However, setting $p_{\mathrm{rand}} = 1$ leads to a model that diverges too fast from the observed sequence.

### A.2   Inference hyper-parameters

#### A.2.1   Image inpainting.

We set the number of epochs to 2, collage frequency to 4 per epoch, top-k sampling to 100, dilation to 1, and number of optimization steps to 200. We apply a gaussian filter on the mask for the periodic image collage.

Additionally, we use these two implementation details. 1) We use two latent masks: the latent down-sampled mask latent_mask$_1$, and the dilated mask latent_mask$_2$, obtained by a dilation of latent_mask$_1$. The randomization is done with latent_mask$_1$, such that no information from the unmasked parts of the image is erased. However, the selection of positions that are re-sampled by EdiBERT is done with latent_mask$_2$. 2) At the second epoch, we randomize the token value, at the position that is being replaced. This is only done for image inpainting.

#### A.2.2   Image compositing.

We set the number of epochs to 2, collage frequency to 4 per epoch, top-k sampling to 100, dilation to 1, and number of optimization steps to 200. We apply a gaussian filter on the mask for the periodic image collage. Contrarily to inpainting, we do not randomize such that EdiBERT samples stay closer to the original sequence.

The full algorithm is presented below in Algorithm 2.

## B   Additional experimental results

We give additional comparisons on FFHQ and LSUN Bedroom, for the following tasks: image inpainting in Table 3, image crossovers in Table 4, and image composition in Table 5. All these experiments are run on the test-set of EdiBERT. Note that concurrent methods based on StyleGAN2 were trained on the full dataset, which advantages them.

---

**Algorithm 2:** Image inpainting/composition

---

**Requires:** Masked (or edited) image $i_m$, mask $m$, Encoder E, Decoder D, BERT model $p_\theta$, epochs
  e, periodic collage c, optimization steps optim_steps;
$s \leftarrow Q(E(i_m))$;
latent_mask $\leftarrow$ get_mask_in_latent_space$(m)$;
**if** *task is inpainting* **then**
  |  $s \leftarrow s \times$ latent_mask $+$ rand $\times$ $(1 -$ latent_mask$)$;
**end**
**for** *e in [0,epochs]* **do**
  |  **for** *p in chosen_order(latent_mask)* **do**
  |    |  Sample token $t \in Z \sim p_\theta^p(\cdot|s)$ ;
  |    |  Insert sampled token: $s_p \leftarrow t$ ;
  |    |  **if** *p%c=0 (collage)* **then**
  |    |    |  Encode image post-collage: $s \leftarrow E(i_m \odot m + D(s) \odot (1 - m))$;
  |    |  **end**
  |  **end**
**end**
$s^0 \leftarrow s$ ;
**for** *i in [0, optim_steps]:* **do**
  |  $L = L_p\big((D(s) - i_m) \odot m\big) + L_p\big((D(s) - D(s^0)) \odot (1 - m)\big)$ ;
  |  $s \leftarrow s + \epsilon * \text{Adam}(\nabla_s L, s)$ ;
**end**
Image $\leftarrow$ Decoder$(s)$;
**Result:** Image

---

**Inpainting.** We use 2500 images. On FFHQ, we provide results for free-form masks and rectangular masks. The height of rectangular masks is drawn uniformly from $[0.4 \times h, 0.6 \times h]$ with $h = 256$, and similarly for the width. For non-rectangular masks generations, we follow the procedure of Chai et al. (2021): we draw a binary mask at low-resolution $6 \times 6$ and uspsample it to $256 \times 256$ with bilinear interpolation.

The ablation study in Table 2 of main paper is performed on free-form masks. Results in Table 1 of main paper are on rectangular masks. On LSUN Bedroom, we provide results for rectangular masks.

**Crossovers.** We generate 2500 crossovers from random pairs of images, on both FFHQ and LSUN Bedroom.

**Editing/Compositing.** We create small datasets of 100 images from the test set of EdiBERT for FFHQ scribble-based editing, FFHQ compositing and LSUN Bedroom compositing. A user study on FFHQ compositing is presented in main paper with statistically significant number of votes. We also provide some metrics in 5. Because of the small size of the dataset, we only report masked L1 and density. For density, the support of the real distribution is estimated with 2500 real points, and density is averaged over the individual density of the 100 generated images.

| | Masked L1 ↓ | FID ↓ | Density ↑ | Coverage ↑ |
|---|---|---|---|---|
| **FFHQ:** rect. masks | | | | |
| I2SG++ (Abdal et al., 2020) | 0.0767 | 23.6 | 0.99 | 0.88 |
| I2SG$^\dagger$++ (Abdal et al., 2020; Karras et al., 2020a) | 0.0763 | 22.1 | **1.25** | 0.91 |
| LC (Chai et al., 2021) | 0.1027 | 27.9 | 1.12 | 0.84 |
| EdiBERT (⋆) | 0.0290 | 13.8 | 1.16 | 0.98 |
| Co-mod. GAN (Zhao et al., 2020) | **0.0128** | **4.7** | 1.24 | **0.99** |
| **FFHQ:** free-form masks | | | | |
| I2SG++ (Abdal et al., 2020) | 0.0440 | 22.3 | 0.92 | 0.89 |
| I2SG$^\dagger$++ (Abdal et al., 2020; Karras et al., 2020a) | 0.0435 | 21.1 | 1.17 | 0.91 |
| LC (Chai et al., 2021) | 0.0620 | 27.9 | 1.22 | 0.85 |
| EdiBERT (⋆) | 0.0201 | 19.4 | 1.14 | 0.96 |
| Com-GAN (Zhao et al., 2020) | **0.0086** | **10.3** | **1.42** | **1.00** |
| **LSUN Bedroom**: rect. masks | | | | |
| I2SG (Abdal et al., 2019) | 0.1125 | 50.2 | 0.04 | 0.04 |
| MaskGIT (Chang et al., 2022) | **0.0209** | 11.4 | **1.09** | 0.96 |
| EdiBERT (⋆) | 0.0288 | **11.3** | 0.89 | **0.97** |

Table 3: Image inpainting.

| | Masked L1 ↓ | FID ↓ | Density ↑ | Coverage ↑ |
|---|---|---|---|---|
| **FFHQ** | | | | |
| I2SG++ (Abdal et al., 2020) | 0.1141 | 29.4 | 0.97 | 0.78 |
| I2SG$^\dagger$++ (Abdal et al., 2020; Karras et al., 2020a) | 0.1133 | 26.9 | **1.35** | 0.82 |
| ID-GAN (Zhu et al., 2020) | 0.0631 | 23.2 | 0.88 | 0.83 |
| LC (Chai et al., 2021) | 0.1491 | 31.9 | 1.17 | 0.77 |
| EdiBERT (⋆) | **0.0364** | **19.7** | 1.05 | **0.88** |
| **LSUN Bedroom** | | | | |
| I2SG (Abdal et al., 2019) | 0.1123 | 45.7 | 0.12 | 0.20 |
| ID-GAN (Zhu et al., 2020) | 0.0682 | 21.4 | 0.35 | 0.57 |
| EdiBERT (⋆) | **0.0369** | **12.4** | **0.64** | **0.84** |

Table 4: Image crossover.

| | Masked L1 $\downarrow$ | Density $\uparrow$ |
|---|---|---|
| **FFHQ scribble-edits** | | |
| I2SG++ (Abdal et al., 2020) | 0.7811 | 0.91 |
| I2SG$^\dagger$++ (Abdal et al., 2020; Karras et al., 2020a) | 0.0777 | 1.11 |
| ID-GAN (Zhu et al., 2020) | 0.0461 | 0.79 |
| LC (Chai et al., 2021) | 0.1016 | **1.14** |
| EdiBERT ($\star$) | **0.0281** | 0.96 |
| **FFHQ compositing** | | |
| I2SG++ (Abdal et al., 2020) | 0.0851 | 0.77 |
| I2SG$^\dagger$++ (Abdal et al., 2020; Karras et al., 2020a) | 0.0866 | **1.07** |
| ID-GAN (Zhu et al., 2020) | 0.0570 | 0.75 |
| LC (Chai et al., 2021) | 0.1116 | 1.00 |
| EdiBERT ($\star$) | **0.0307** | 0.94 |
| **LSUN Bedroom compositing** | | |
| I2SG (Abdal et al., 2019) | 0.1285 | 0.25 |
| ID-GAN (Zhu et al., 2020) | 0.0484 | 1.45 |
| EdiBERT ($\star$) | **0.0247** | **1.49** |

Table 5: Image editing.

## C    Baselines

We use the implementation and pre-trained models from the following repositories.

ID-GAN (Zhu et al., 2020): https://github.com/genforce/idinvert_pytorch, which has pre-trained models on FFHQ 256x256 and LSUN Bedroom 256x256.

I2SG++ and I2SG†++(Karras et al., 2020b;a; Abdal et al., 2020): https://github.com/NVlabs/stylegan2-ada-pytorch. We tested projections with the following pre-trained models on FFHQ: StyleGAN2 (Karras et al., 2020b) at resolution 256x256, and StyleGAN2-Ada (Karras et al., 2020a) at resolution FFHQ 1024x1024. For evaluation, we downsample the 1024x1024 generated images to 256x256.

LC (Chai et al., 2021): https://github.com/chail/latent-composition. We use the pre-trained encoder and StyleGAN2 generator, for FFHQ at resolution 1024x1024. For evaluation, we downsample the 1024x1024 generated images to 256x256.

Com-GAN Zhao et al. (2020): https://github.com/zsyzzsoft/co-mod-gan. We use the pre-trained network for image inpainting on FFHQ at resolution 512x512. We downsample the generated images to 256x256 for evaluation.

MaskGIT (Chang et al., 2022): https://github.com/google-research/maskgit. We use the tokenizer and transformer trained for conditional image generation and editing on ImageNet 256x256. To perform comparisons with EdiBERT on LSUN Bedroom Image Inpainting, we condition the transformer to the ImageNet class '843: studio couch, day bed'.

## D    Qualitative results on image compositon

We present more examples of image compositions, with image compositing and scribble-based editing on FFHQ and LSUN Bedroom in Figure 9, 10, and 11.

**Preservation of non-masked parts.** Thanks to its VQGAN auto-encoder, EdiBERT generally better conserves areas outside the mask than GANs inversion methods. This is particularly visible for images with complex backgrounds on FFHQ (Figure 10, 5th and last rows).

**Insertion of edited parts.** Since EdiBERT is a probabilistic model and the tokens inside the modified area are resampled, the inserted object can be modified and mapped to a more likely object given the context. It thus generates more realistic images, but can alter the fidelity to the inserted object. For example, on row 1 of Figure 11, the green becomes lighter and the perspective of the inserted window is improved. Although it can be a downside for image compositing, note that this property is interesting for scribble-based editing, where the scribbles have to be largely transformed to get a realistic image. Contrarily, GANs inversion methods tend to conserve the inserted object too much, even if it results in a highly unrealistic generated image. We can observe this phenomenon on last row of Figure 10.

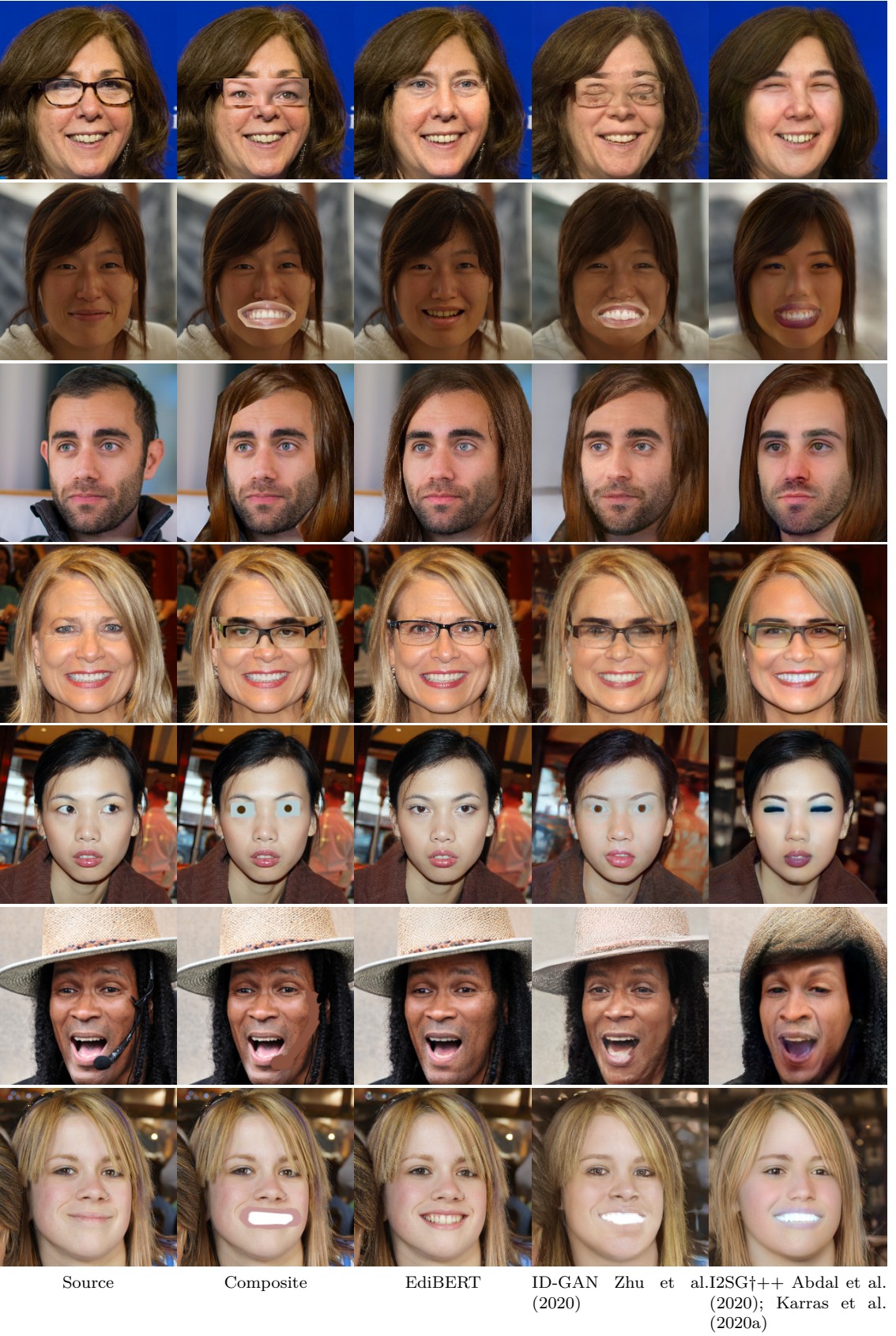

| Source | Composite | EdiBERT | ID-GAN Zhu et al. (2020) | I2SG†++ Abdal et al. (2020); Karras et al. (2020a) |

Figure 10: Image compositing and scribble-based editing on FFHQ.

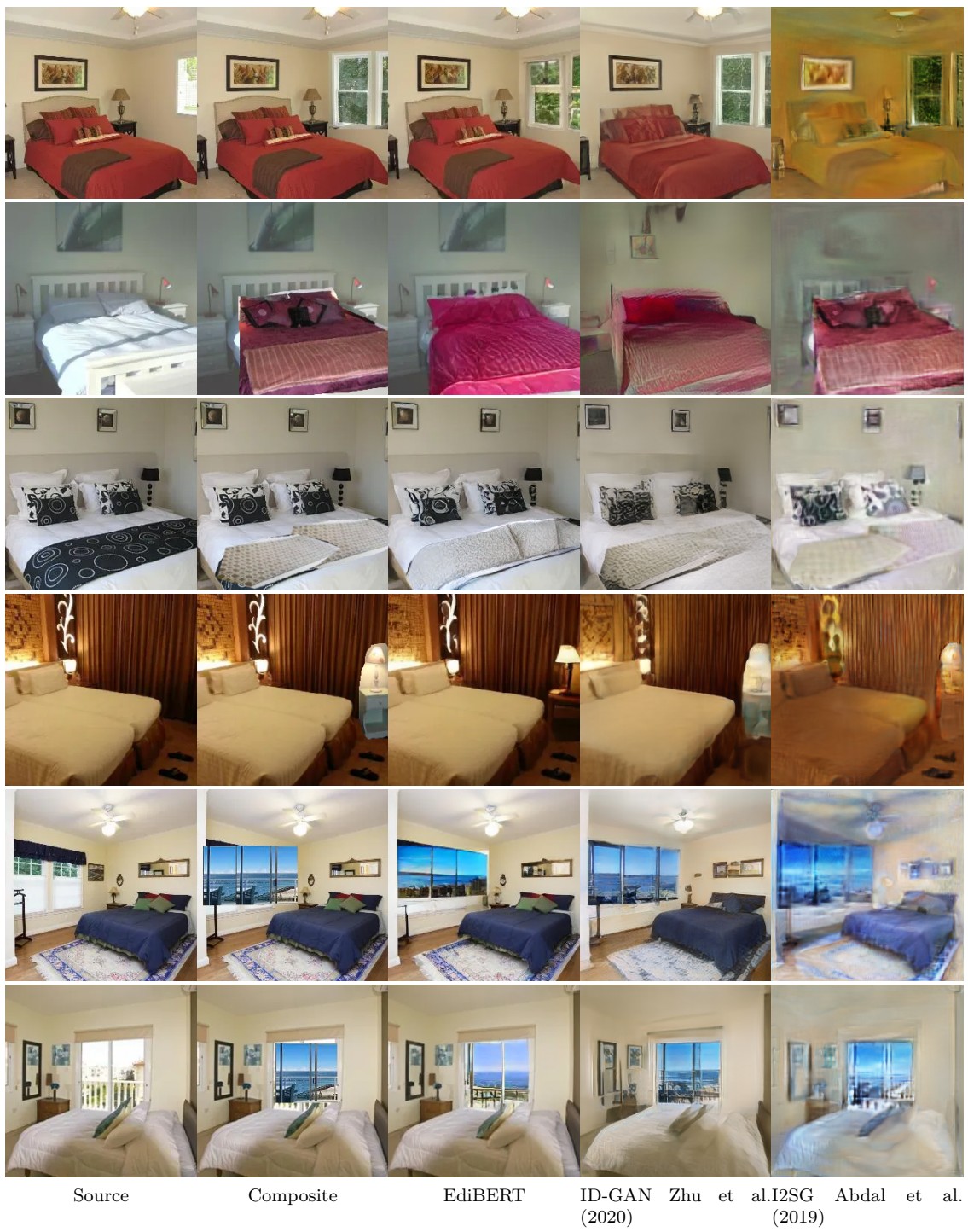

Source          Composite          EdiBERT          ID-GAN Zhu et al.  I2SG Abdal et al.
                                                     (2020)            (2019)

Figure 11: Image compositing on LSUN Bedroom.

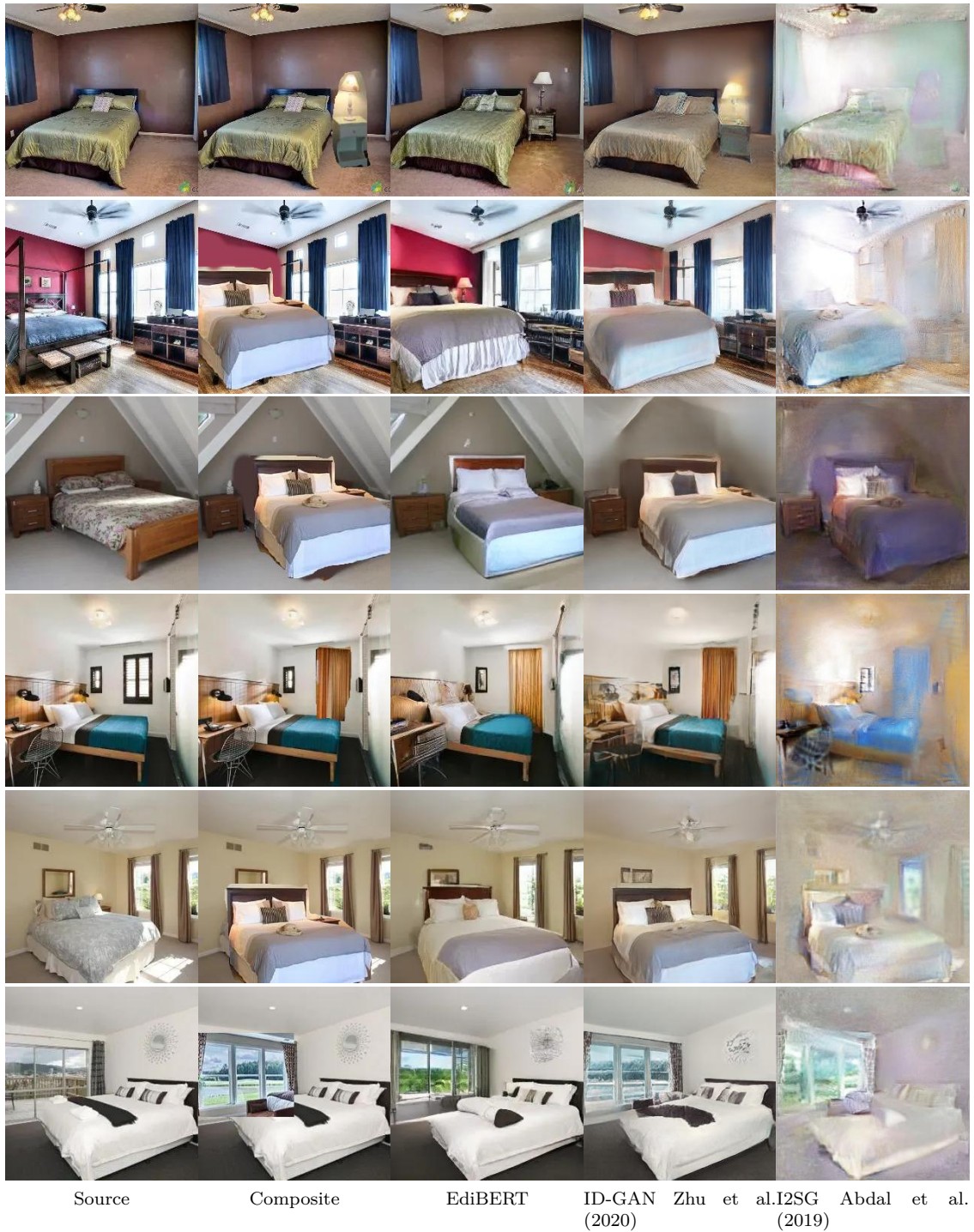

Source      Composite      EdiBERT      ID-GAN Zhu et al. (2020)      I2SG Abdal et al. (2019)

Figure 12: Image compositing on LSUN Bedroom.

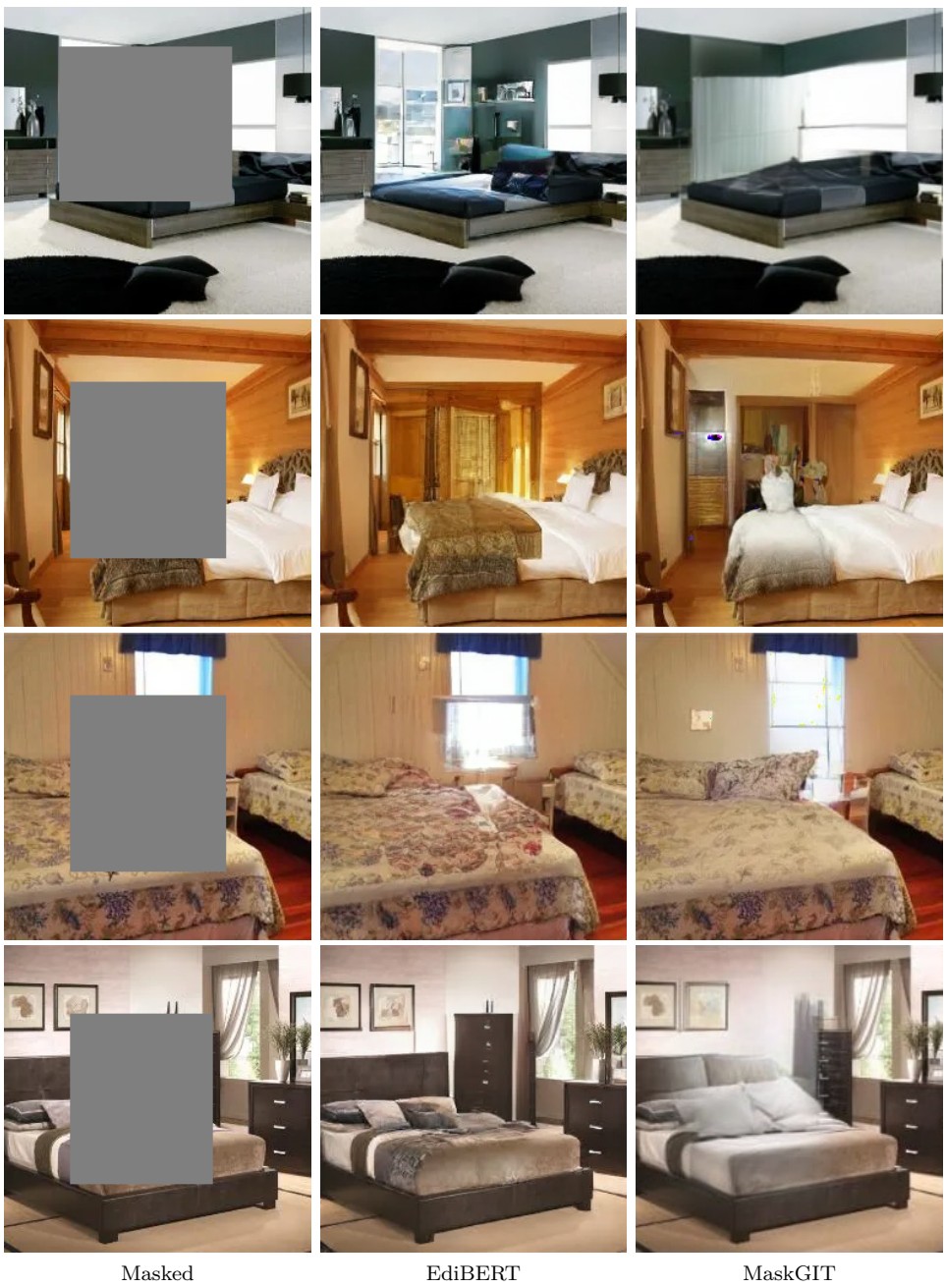

Masked             EdiBERT             MaskGIT

Figure 13: Comparisons of image inpainting on LSUN Bedroom between EdiBERT and MaskGIT.

# E   Survey on FFHQ image compositing

The survey was presented as a Google Form with 40 questions. For each question, the user was shown 6 images: Source, Composite, EdiBERT, ID-GAN Zhu et al. (2020), I2SG†++ Abdal et al. (2019); Karras et al. (2020a), LC Chai et al. (2021). The different generated images were referred as Algorithm 1, ..., Algorithm 4. The user was asked to vote for its preferred generated image, by taking into account realism and fidelity criterions. The user had no time limit for the poll. 30 users answered our poll. We provide the detailed answers for each image in Table 6.

| | EdiBERT | ID-GAN Zhu et al. (2020) | LC Chai et al. (2021) | I2SG†++ Abdal et al. (2019); Karras et al. (2020a) |
|---|---|---|---|---|
| | 17 | 5 | 7 | 1 |
| | 15 | 1 | 8 | 6 |
| | 22 | 4 | 1 | 3 |
| | 19 | 4 | 5 | 2 |
| | 22 | 0 | 4 | 4 |
| | 6 | 7 | 8 | 9 |
| | 21 | 1 | 5 | 3 |
| | 23 | 1 | 4 | 2 |
| | 20 | 5 | 5 | 0 |
| | 11 | 13 | 6 | 0 |
| | 27 | 0 | 3 | 0 |
| | 12 | 3 | 3 | 12 |
| | 16 | 4 | 6 | 4 |
| | 25 | 2 | 1 | 2 |
| | 18 | 8 | 1 | 3 |
| | 8 | 13 | 9 | 0 |
| | 26 | 0 | 4 | 0 |
| | 7 | 0 | 21 | 2 |
| | 14 | 9 | 1 | 6 |
| | 27 | 0 | 1 | 2 |
| | 11 | 19 | 0 | 0 |
| | 14 | 9 | 4 | 3 |
| | 16 | 14 | 0 | 0 |
| | 21 | 1 | 3 | 5 |
| | 8 | 2 | 18 | 2 |
| | 19 | 3 | 3 | 5 |
| | 22 | 7 | 0 | 1 |
| | 23 | 2 | 1 | 4 |
| | 18 | 0 | 2 | 10 |
| | 27 | 2 | 1 | 0 |
| | 22 | 2 | 1 | 5 |
| | 24 | 0 | 5 | 1 |
| | 3 | 25 | 2 | 0 |
| | 28 | 0 | 2 | 0 |
| | 24 | 0 | 6 | 0 |
| | 27 | 0 | 3 | 0 |
| | 27 | 1 | 2 | 0 |
| | 22 | 7 | 1 | 0 |
| | 9 | 15 | 6 | 0 |
| | 14 | 0 | 14 | 2 |
| Total | 735 (61.25%) | 189 (15.75%) | 177 (14.75%) | 99 (0.0825%) |

Table 6: Detailed results of the user study. Each line corresponds to an image, with the associated number of votes per method.

