# OpenReview forum: "EdiBERT: a generative model for image editing"
_TMLR — Accepted by TMLR_

### Review · Reviewer_DPNU · 2022-08-24

**Summary Of Contributions:**

In this work the authors propose a method to perform image edition/manipulation using VQ-GAN and attention models.

Starting from the VQ-GAN model with its tokens, an image can be represented as a sequence of such tokens. During training, the sequence of tokens is randomly perturbed and the model is trained to recover the original sequence from the full perturbed input sequence. This objective is different from the more commonly used autoregressive model for unconditional image generation.

After training the model, the authors analyze the effect of image space manipulation on the latent space. Another aspect that needs to be addressed is the reconstruction capabilities. Using the default token sequence estimation results in images that are further away from the input. To address this the authors use latent space optimization with the objective of reconstructing the input image.

After all this, the image edition/manipulation is expressed as an optimization problem over the sequence of tokens. The exact steps and terms of this optimization vary depending on the exact problem (denoising vs inpainting).


**Broader Impact Concerns:**

The authors acknowledge that the model might be used to create and propagate fake beliefs via deepfakes, a concern shared with other image generative models.

**Requested Changes:**

Currently I would recommend the paper for acceptance. The necessary changes are:
1. More details about the model, in particular the different distributions and the attention mechanism.
2. Visual example with other state of the art methods focusing on reconstruction gap
3. Visual comparisons with existing methods on other tasks should appear in the main paper.

Addressing the remaining points from the weaknesses section would strengthen the paper.


**Strengths And Weaknesses:**

# Strengths

- Use VQ-GAN for image denoising and manipulation
- Quantitative evaluation shows competitive performance and user study gives the proposed method a clear advantage.
- Overall the paper is clear and easy to follow
- The ablation study shows the necessity of the different tricks proposed by the authors to achieve best results and avoid divergence from the input image.

# Weaknesses

1. The presentation of the model assumes the reader knows in detail of the VQ-GAN (Esser et al. 2021b). Not enough details are provided for the different distributions and attention mechanisms. This is important to judge on the novelty of the proposed model.
2. Although the reconstruction is improved thanks to the latent space optimization, there is still an important remaining gap.
3. Currently the proposed optimization algorithm relies on several tricks and guides to avoid divergence. It would be interesting to have some more comments on how feasible it would be to shape the latent space to render those tricks unnecessary.
3. Limited visual comparisons: only the composition task has images and the images are provided in supplemental.
4. Exploring an additional dataset would be interesting. What is the expected performance for a more general setting (beyond faces or interior scenes)?

---

### Review · Reviewer_ucMa · 2022-08-30

**Summary Of Contributions:**

In this paper, the authors propose a generative model for image editing. It is claimed that a single pretrained model could be applied to image denoising, image inpainting, and image composition tasks. The problem of a previous method VQGAN is analyzed, which helps to identify the chance to improve that model. In particular, an additional optimization step is incorporated, namely, optimizing the LPIPS metric over the latent spaces. It is visualized that by adding this additional optimization step, more high-frequency details are recovered. Then the authors present how to adapt one single model to two different image manipulation tasks, namely image denoising and image inpainting. It is also show that competitive results can be achieved with the generic training algorithm.

**Broader Impact Concerns:**

The authors base their experiments on exsiting public datasets. Thus, although the experimental results is conducted on human faces, there is no ethical issues at lease for the algorithm part.

**Requested Changes:**

Questions:
1. The authors mentioned that "An autoregressive model is trained to generate these token sequences directly, stressing that high-capacity
expressive transformers can generate realistic high-resolution images. To summarize, this framework consists of three main steps:" Could the authors explain the difference between the three main steps mention here and the main steps used by the proposed method?
2. Please revised the paper according to the suggestions given above. Please make the figures and contents more self-contained. Please explain each equation more clearly.

**Strengths And Weaknesses:**

Strengths:
1. It is a good idea to design a single model for image manipulation tasks. Unified models are already proven to be useful for language tasks. And for high-level vision tasks, CLIP models are shown to generalize quite well to images with domain gap. This paper is inspired by the general idea and tries to propose a unified approach for image manipulation.
2. According to experimental results, the proposed method achieves comparable subjective and objective performances, although only a single pretrained model is used.

Weakness:
1. The main contribution of this paper is the change of the optimization method of VQGAN, that is incorporating an additional optimization LPIPS loss over the latent space. The main contribution is a little bit trivial.
2. The paper is not well-written. And a lot of desciptions are not self-contained. And a couple of them are listed below.
    - The pipeline of the proposed framework is not clearly presented in the begining of this paper. A figure that describes the general pipeline would help to improve the readability of this paper.
    - To adapt a single pretrained network to two different image manipulation task including denoising and image inpainting, the resampling strategy seems to be essential. But it is not explained well in the introduction why this resampling strategy is important.
    - The equations and notations are not well explained. For example, what does $E(I)_{l}$ in Eqn. 1 mean? Only after reading another paper did I undertand the correct meaning. Again, what does the $Z^l$ in the following line mean?
    - Some description is not self-contained. For example, after Eqn. 6, the authors mentioned that "the sampling function φ is defined with a 2D masking strategy, and the training positions are selected by drawing random 2D rectangles in the 2D patch extracted from the encoder." Please explain explicitly what the meaning of 2D masking strategy is.
    - Figures needs to be well explained. For example, in Figure 2: Modifying the image via the latent space, what does the four images mean? How are the third and fourth images formed is not explained.

[1] Taming Transformers for High-Resolution Image Synthesis

---

> ### Author Response · Authors · 2022-09-08
> **Response to the review of Paper299 by Reviewer ucMa**
>
> First, we would like to thank you for the time dedicated to review our paper and for your insightful comments, which allowed us to clarify our setting. Please find responses to your concerns below. Also, note that we uploaded a revised version of our paper.
>
> **The main contribution of this paper is the change of the optimization method of VQGAN, that is incorporating an additional optimization LPIPS loss over the latent space.**
>
> We want to highlight that another main contribution is the use of bidirectional transformers for image manipulation. Autoregressive transformers rely on log-likelihood training objective and left-to-right sampling algorithm. The use of bidirectional transformers requires both different training objectives and sampling algorithms. If we used a standard training objective for bidirectional transformers (BERT-like), we carefully designed the different sampling algorithm and validated our choices in an ablation study.
>
> **To adapt a single pretrained network to two different image manipulation task including denoising and image inpainting, the resampling strategy seems to be essential. But it is not explained well in the introduction why this resampling strategy is important.**
>
> Agreed, there are some differences but both sampling algorithms for inpainting and denoising are overall quite close: both rely on sampling tokens from the output distributions of EdiBERT. The main difference is how one selects the positions: for denoising, one does not know *a priori* the noisy positions (to be edited), while for inpainting, positions to sample are known through a pixel-space mask. A second difference is that inpainting requires a collage of the pixels outside the mask. We made it clearer in the introduction that EdiBERT was based on a single training but required two different sampling algirthms.
>
> **The equations and notations are not well explained. For example, what does $E(I)_l$ in Eqn. 1 mean? Only after reading another paper did I undertand the correct meaning. Again, what does the $Z^L$ in the following line mean?**
>
> Thank you for this remark, we amended the paper accordingly. We detailed the meaning of $E(I)_l$ in the paper: it corresponds to the feature vector of $E(I)$ at position $l$. For $s \in Z^L$, it means that $s$ is a set of $L$ elements that belong to $Z$.
>
> **Some description is not self-contained. For example, after Eqn. 6, the authors mentioned that "the sampling function φ is defined with a 2D masking strategy, and the training positions are selected by drawing random 2D rectangles in the 2D patch extracted from the encoder." Please explain explicitly what the meaning of 2D masking strategy is.**
>
> Thank you for this remark. We made this clearer in the paper and added Figure 2 to explicitly describe the 2D selection strategy.
>
> **Figures needs to be well explained. For example, in Figure 2: Modifying the image via the latent space, what does the four images mean? How are the third and fourth images formed is not explained.**
>
> We agree that an improvement could be done on the descriptions of the different figures. All captions have been updated and significantly improved.
>
> **The authors mentioned that "An autoregressive model is trained to generate these token sequences directly, stressing that high-capacity expressive transformers can generate realistic high-resolution images. To summarize, this framework consists of three main steps:" Could the authors explain the difference between the three main steps mention here and the main steps used by the proposed method?**
>
> Our work directly builds on top of the previous works of VQVAE and VQGAN: the first step where the triplet (Encoder, Decoder and Codebook) is trained, does not change (this is now better explained in Related Work). However, steps 2 and 3 are modified.
>
> In 2: instead of an autoregressive transformer, we train a bidirectional transformer with a BERT-like training objective. This is explained in Section 3.3.
>
> In 3: we do not sample tokens in an auto-regressive manner (ancestral sampling) at inference time. Instead, we derived sampling algorithms designed for image manipulation in Section 4 (see Algorithm 1 and 2 for more details).

---

### Review · Reviewer_ANEH · 2022-09-13

**Summary Of Contributions:**

This paper presents EdiBERT, a bidirectional transformer-based model trained with a simple masked token prediction task in the discrete latent space of a VQGAN and capable of inpainting, scribble-based image processing, and image harmonization/composition. Particular emphasis is placed on the ability to achieve these functions with a single model. The model is trained on 256x256 images of the FFHQ and LSUN datasets and achieves results that rival other more specialized models.


**Broader Impact Concerns:**

Ethical implications of the model are briefly mentioned. While the discussion is quite brief and only mentions deepfakes as a potential negative impact, a full discussion of the societal implications of powerful generative image models is likely beyond the scope of this paper and an ongoing community effort.

**Requested Changes:**

Detailed requests/clarifications can be found in the **weaknesses** section above. In general, I think that the paper should address more the possible shortcomings and discuss the related work mentioned above. I am also somewhat concerned about the applicability/relevance of the proposed approach. It seems to be limited to "simple datasets" such as FFHQ and LSUN and does not provide a clear advantage over other approaches. The more general discrete diffusion models seem more promising in this regard (while also providing a "unified" model), and the proposed approach could be interpreted as a particular denoising sampling scheme.


**Strengths And Weaknesses:**

**Strengths:**

The paper is well written and easy to follow, the proposed approach is simple. Further, the manuscript does a good job of motivating a bidirectional approach over AR models for image manipulation tasks. Interesting analysis of the latent space of a f16-VQGAN are presented.

**Weaknesses:**

- The proposed approach is effectively making an independence assumption for the masked tokens. The discussion on where this fails too short (only reference I can find is on p.14, "However, some manipulations such as zooms or rotations require changing large areas of the source image. In these cases, modifying a large number of tokens might be more demanding."
- Further, the paper builds on AR factorization as a motivation to introduce bidirectional attention, i.e. a generative model that learns a "full" density, but itself presents an approach that makes a strong assumption on the density (i.e. independence of individual tokens). Again, advantages and disadvantages of such a modeling choice should be discussed (e.g., trading speed and control vs generative capabilities of complex content?). I also suspect that the approach will not work as well for larger masks, more complex scenes/datasets and VQ-Models with a downsampling factor of 8 oder 4 instead of 16.
- Related: The paper should compare to approaches that use bidirectional models in latent space, most notably MaskGIT [1] and M6 [2] and to ones that build discrete diffusion [3, 4] models in latent space [5]. Furthermore, a discrete diffusion formulation formulation is highly related to the one described here and opens up a different view on the approach as a few-step decoding sampler of a multinomial diffusion process. The only reference to diffusion models is to SDEdit with a reference to decoding speed.
- A discussion/ablation on the number of inference iterations T is missing.
- The results are focused on FFHQ, a simple and highly structured dataset. How does the inpainting approach on more complex datasets. A common benchmark for inpainting models (such as the mentioned CoModGAN) is the Places dataset [6]
missing evaluation of applying the method in different latent spaces. E.g., is it possible to train an EdiBERT in an f=8 latent space?

**Minor remarks/Questions:**

- p.18, A.1. "In our experiments, since we work at resolution 256 × 256 and follow the downsampling factor of 4 from Esser et al. (2021b), we have h = w = 256/4 = 16." --> My understanding from the main paper is that the downsampling factor is 16.
- it would be nice to see more qualitative results on LSUN-Bedrooms, especially for inpainting.
- a comparison to other models in terms of sampling speed is missing. While the approach is significantly faster than AR models, it aims to fulfill a different tasks (image manipulation vs full generation) and should thus be compared to related approaches. How expensive is the "post-collage" procedure described in Alg.2? How much quality is lost without post-hoc latent optimiation, and how much speed gained?
- p.1: litterature -> literature


_Refererences_
- [1]: MaskGIT: Masked Generative Image Transformer, Chang et al, https://arxiv.org/abs/2202.04200
- [2]: M6-UFC: Unifying Multi-Modal Controls for Conditional Image Synthesis via Non-Autoregressive Generative Transformers, Zhang et al, https://arxiv.org/abs/2105.14211
- [3]: Argmax Flows and Multinomial Diffusion: Learning Categorical Distributions, Hoogeboom et al, https://arxiv.org/abs/2102.05379
- [4]: Structured Denoising Diffusion Models in Discrete State-Spaces, Austin et al, https://arxiv.org/abs/2107.03006
- [5]: Vector Quantized Diffusion Model for Text-to-Image Synthesis, Gu et al, https://arxiv.org/abs/2111.14822
- [6]: Places: A 10 million Image Database for Scene Recognition, B. Zhou, A. Lapedriza, A. Khosla, A. Oliva, and A. Torralba, IEEE Transactions on Pattern Analysis and Machine Intelligence, 2017

---

> ### Author Response · Authors · 2022-09-16
> **Response of review of Paper299 by Reviewer ANEH (1)**
>
> Thank you for the time dedicated to review our paper, and for your insightful comments. Please find detailed responses to your concerns below. Also, note that we uploaded a revised version of our paper.
>
> **Masking strategy: The proposed approach is effectively making an independence assumption for the masked tokens. The discussion on where the proposed approach fails is too short.**
>
> Indeed, our approach makes an independence assumption during training time. However, note that this independence assumption is only done between the perturbed tokens. Besides, this independence assumption is not used at inference time, since tokens are replaced one by one conditionally to the whole sequence. Interestingly, we found out that MaskGIT[1] also uses an independence assumption on the tokens both at training and inference time. Using this training and sampling strategy, they reach state-of-the-art results in unconditional image generation.
>
> **Bidirectionnal vs auto-regressive modeling: advantages and disadvantages of such a modeling choice should be discussed (e.g., trading speed and control vs generative capabilities of complex content?).**
>
> On one hand, auto-regressive models learn densities of the sequence and are theoretically better suited for sequence generation. For example, the latest autoregressive PARTI model has shown impressive results in text-to-image generation. However, these models are inappropriate for image manipulation tasks, or require long sampling time when used in a denoising setting: the autoregressive model ImageBART needs to generate full sequences several times for one single image manipulation. On the other hand, we show in this paper that bidirectional models are powerful and efficient for localized image manipulations. Moreover, since they are not tied to a specific sequence ordering, they allow for adapted sampling procedures.
>
> **Bidirectionnal SOTA: The paper should compare to approaches that use bidirectional models in latent space, most notably MaskGIT [1] and M6 [2] and to ones that build discrete diffusion [3, 4] models in latent space [5]. The only reference to diffusion models is to SDEdit with a reference to decoding speed.**
>
> The main difference between us and these other VQ based bi-directionnal models [1] and [2] is the masking strategy. While these other methods use MASK tokens, we directly perturb the sequence of tokens. We argue that this is important for tasks such as scribble-based editing or denoising, where all the information given by the user should be used. Regarding the literature on diffusion models, we mentioned both SDEdit, a continuous-space diffusion model, and ImageBART, a discrete diffusion model building on [3], and have now updated the related work in line with your suggestion.
>
> **Downsampling factor: I also suspect that the approach will not work as well for larger masks, more complex scenes/datasets and VQ-Models with a downsampling factor of 8 oder 4 instead of 16. Is it possible to train an EdiBERT in an f=8 latent space?**
>
> Indeed, the downsampling factor used is 16, and this typo has been corrected. A lower downsampling factor, such as 8, would be computationally more involved since the transformer would need to handle sequences of length $32\times32=1024$. We believe our method would scale to a lower downsampling factor, and our intuition is that an important hyper-parameter would be the size and diversity of the rectangle masks during training.
>
> **More complex datasets: How does the inpainting approach on more complex datasets. A common benchmark for inpainting models (such as the mentioned CoModGAN) is the Places dataset [6]. (also) It would be nice to see more qualitative results on LSUN-Bedrooms.**
>
> Since we do not use any specific method for the two tested datasets, we expect good performances on a more general setting and also more complex datasets. Unfortunately, we could not test EdiBERT on these much larger datasets for limited computational reasons. For example, to train on the LSUN Bedroom dataset, with 3 million images, we have used 8 Tesla V100 GPUs with 32Gb. The VQGAN was trained for 3 days, the training of the transformer was stopped after for 5 days, even though the models did not reach saturation point (validation error still improving). We have added a new image in Appendix showing supplementary results regarding the compositing task: on the newly added results, EdiBERT qualitatively outperforms the competition.

---

> > ### Author Response · Authors · 2022-09-16
> > **Response of review of Paper299 by Reviewer ANEH (2)**
> >
> > **Sampling speed: While the approach is significantly faster than AR models, it [...] should thus be compared to related approaches. How expensive is the "post-collage" procedure [..]? How much quality is lost without post-hoc latent optimization, and how much speed gained?**
> >
> > While a simple sampling procedure with EdiBERT would take around 6 seconds (with a Tesla V100 GPU), the whole "post-collage" optimization (that requires both the encoder and the decoder) takes around 20 seconds. For comparison, IdGAN takes around 20 seconds and Image2StyleGAN++ around 1 minute. Figure 8 and Table 2 show that post-collage improves fidelity to the source image. Finally, with respect to post-hoc latent optimization, we have added a new Figure 4 that stresses that the reconstruction abilities of EdiBERT outperforms the competition after the procedure.

---

### Decision · Action_Editors · 2022-11-29

**Recommendation:** Accept with minor revision

**Comment:**

The paper presents a bidirectional transformer that re-samples image patches conditioned on an input image. The main advantage of the proposed method is that a *single* trained model can be applied to solve a wide variety of different image editing tasks.

Reviewer ucMa and Reviewer DPNU are satisfied with the authors' detailed revision and response, including a discussion on missed references in the related work section and various clarifications on method exposition. After the discussions with the authors, both Reviewer ucMa and Reviewer DPNU are leaning toward acceptance. Reviewer ANEH, however, is leaning toward rejection. The main complaints were a lack of comparisons with MaskGIT and M6-UFC, simple, structured datasets (e.g., FFHQ), and some method exposition issues.

The authors' responses partially address these concerns, including the revision of the related work to include relevant prior work and an explanation of the lack of other larger-scale datasets. The AE thinks the responses and revision sufficiently address the concerns of Reviewer ANEH. That being said, the AE agrees with Reviewer ANEH that a comparison with methods that also use bidirectional models in latent space, e.g,. MaskGIT and M6-UFC, would be beneficial for the readers to get a clearer picture of the literature. The AE thus recommends "Accept with minor revision".

Specifically, the authors should
1) cite and discuss the differences between the proposed method and MaskGIT and M6-UFC
2) include at least one direct comparison with MaskGIT (code available here: https://github.com/google-research/maskgit).

Note that the acceptance is *not* tied to the performance comparison with MaskGIT. The MaskGIT requires training one specific model per task so there are still merits of the proposed method even if it underperforms MaskGIT on the compared tasks.

**Audience:**

The computer vision community will likely be interested in learning this work on its unified approach to image editing.

**Claims And Evidence:**

The claims made in the submission --- a unified approach for image editing --- are accurate and supported by an extensive set of experiments.

---

> ### Author Response · Authors · 2022-12-13
> **Response to AE**
>
> We thank both the AE and the reviewers for their time and their insightful remarks regarding our work. Following both the AE’s and Reviewer ANEH’s suggestions, we added new comparisons with MaskGit on the inpainting task. On the LSUN Bedroom dataset, we compared qualitatively and quantitatively EdiBERT with MaskGit (see Table 3 and Figure 13 in Appendix). On the few sampled examples, both models perform similarly. However, we observe that MasktGit slightly outperforms EdiBERT on the L1 reconstruction and density metrics. As pointed out by AE, this was expected since MaskGit uses [MASK] tokens, and is thus better suited for inpainting. The new version of the paper has now been uploaded.